# DHX9 SUMOylation is required for the suppression of R-loop-associated genome instability

Bing-Ze Yang[1,3], Mei-Yin Liu[1,3], Kuan-Lin Chiu[2], Yuh-Ling Chien[1], Ching-An Cheng[1], Yu-Lin Chen[1], Li-Yu Tsui[1], Keng-Ru Lin[1], Hsueh-Ping Catherine Chu[2] & Ching-Shyi Peter Wu[1] ✉

RNA helicase DHX9 is essential for genome stability by resolving aberrant R-loops. However, its regulatory mechanisms remain unclear. Here we show that SUMOylation at lysine 120 (K120) is crucial for DHX9 function. Preventing SUMOylation at K120 leads to R-loop dysregulation, increased DNA damage, and cell death. Cells expressing DHX9 K120R mutant which cannot be SUMOylated are more sensitive to genotoxic agents and this sensitivity is mitigated by RNase H overexpression. Unlike the mutant, wild-type DHX9 interacts with R-loop-associated proteins such as PARP1 and DDX21 via SUMO-interacting motifs. Fusion of SUMO2 to the DHX9 K120R mutant enhances its association with these proteins, reduces R-loop accumulation, and alleviates survival defects of DHX9 K120R. Our findings highlight the critical role of DHX9 SUMOylation in maintaining genome stability by regulating protein interactions necessary for R-loop balance.

During transcription, nascent RNA strongly binds to the DNA template, creating a transient DNA/RNA hybrid and a displaced single-stranded DNA (ssDNA) on the non-template strand. This structure, known as an R-loop, occurs naturally from bacteria to humans and plays a crucial role in cellular physiology[1–6]. For instance, in human B cells, R-loops are pivotal for immunoglobulin class switching at switch regions[7]. Furthermore, the distribution of R-loops across the genome has been mapped[8–10], with over half preferentially binding to promoter regions, where R-loops can influence gene expression. Particularly, R-loops are prevalent near promoter regions exhibiting a high G/C skew, which prevents the binding of DNA methyltransferases to these regions for DNA methylation[10]. Additionally, R-loops are abundant near the 3′-end regions of polyA-dependent genes, contributing to efficient transcriptional termination[9,11]. Despite their significance in cellular processes, dysregulated R-loops arising from RNA-processing defects are implicated in human diseases, such as neurological disorders and cancer[5,6].

Several distinct mechanisms crucial for modulating the R-loop levels within cells have been unveiled[12–14]. RNase H (RNH) is an enzyme responsible for the degradation of the RNA moiety in a DNA/RNA hybrid[15]. In human cells, loss of RNH1 results in increased DNA double-stranded breaks (DSBs) and genome instability. Conversely, over-expression of RNH1 effectively mitigates the accumulation of R-loop-associated genome instability in RNA-processing-deficient cells[13,16]. Proteins like BRCA1, BRCA2, FANCD2, and FANCM, involved in the DNA damage response (DDR) are also involved in suppressing unprogrammed R-loops[17–21]. Additionally, helicases Senataxin (SETX) and Aquarius (AQR), which respectively participate in promoting transcriptional termination and DNA/RNA hybrid resolution, play crucial roles in suppressing aberrant R-loops[11,22,23]. Interestingly, despite their distinct biological functions in RNA metabolism, a group of DExD/H-box RNA helicases, including DDX1, DDX5, DDX17, DDX18, DDX19, DDX21, DDX41, and DHX9, exhibit critical functions in modulating R-loop homeostasis[24–31]. It remains unclear whether RNA

[1]Department and Graduate Institute of Pharmacology, College of Medicine, National Taiwan University, Taipei 100233, Taiwan. [2]Institute of Molecular and Cellular Biology, National Taiwan University, Taipei 106319, Taiwan. [3]These authors contributed equally: Bing-Ze Yang, Mei-Yin Liu. ✉e-mail: cswu2017@ntu.edu.tw

helicases function individually at specific loci or collaborate to efficiently resolve unscheduled R-loops in cells with impaired RNA processing.

Initially known as DNA helicase II and RNA helicase A[32], DHX9 is part of the RNA polymerase II (Pol II) holoenzyme, playing a crucial role in co-transcriptional pre-mRNA processing[33]. With its helicase activity, DHX9 resolves abnormal secondary structures of nucleic acid[34], including DNA/RNA hybrids and DNA/RNA guanine quadruplexes (G4-DNA/RNA), potential sources of genomic instability. Emerging evidence has shed light on the tight correlation between dysregulated DHX9 and various types of cancer[32]. Furthermore, certain monoallelic variations in DHX9 have been linked to neuronal disorders[35,36]. Despite its significance in R-loop homeostasis[26], the precise regulatory mechanisms governing DHX9 in suppressing R-loops during transcription and replication conflicts remain incompletely understood. SUMOylation involves covalent conjugation of small ubiquitin-like modifier(s) (SUMO), diversifies protein function by changing subcellular localization, affecting protein stability, and altering protein-protein interactions[37]. In mammals, three major paralogues, SUMO1-3, have been extensively investigated. While the sequences of SUMO2 and SUMO3 are almost identical, SUMO1 shares only ~50% similarity with SUMO2/3. The process of attaching SUMO to a target protein begins with SUMO activation by an E1 activating enzyme, followed by transfer to an E2 conjugating enzyme, UBC9, which then conveys SUMO to the target protein, facilitated by an E3 ligase. Over one-third of SUMOylation sites are found in a consensus SUMOylation motif, $\psi KxE/D$, where $\psi$ is a hydrophobic amino acid, and $x$ is an arbitrary amino acid[38]. Besides SUMOylation, numerous proteins bind non-covalently to SUMO via SUMO-interacting motifs (SIMs), influencing a range of cellular processes[37,38]. Recent studies have revealed the regulatory impact of SUMOylation on RNA metabolism[39]. For instance, SUMOylation of spliceosomal proteins enhances pre-mRNA splicing efficiency[40]. SUMOylation of SETX directs exosomes to transcription-induced DNA damage sites, and mutations that compromise SETX SUMOylation are linked to certain neurological diseases[41]. DHX9 has been described to interact with SUMO E2 conjugase UBC9 via its N-terminal amino acid residues 1–137 and can be conjugated by SUMO1[42]. Interestingly, system-wide studies on endogenous SUMO2 conjugation sites reveal that DHX9 is modified by SUMO2. Nevertheless, the SUMO2 conjugation sites mapped in these studies exhibit inconsistencies[43,44]. Therefore, the specific residue(s) on DHX9 that undergo SUMOylation in cells and the function of DHX9 SUMOylation remain uncertain.

In this study, we focus on understanding the impact of DHX9 SUMOylation on R-loop regulation. We confirm that SUMOylation of DHX9 by SUMO2/3 at K120 is crucial for modulating R-loop balance. Through SUMO-SIM interactions, DHX9 SUMOylation enhances its association with multiple R-loop interacting proteins, including PARP1 and DDX21, thereby modulating R-loop dynamics and safeguarding genome stability.

## Results

### K120 is a major SUMO conjugation site of DHX9
Through extensive examination of previous system-wide studies on endogenous protein SUMOylation, we uncovered DHX9 SUMOylation in two separate studies[43,44]. While one study mapped K5, K76, K120, and K275 as SUMO2 acceptor sites, the other identified K697 (Fig. 1a, b). This discrepancy led us to use two distinct computational programs, GPS-SUMO[38] and SUMOplot analysis program, to predict potential DHX9 SUMOylation sites. Among the five lysine residues identified by proteomic studies, K76 and K120, located in different consensus SUMOylation motifs, garnered high scores in both computational programs (Supplementary Fig. 1a, b). Thus, we decided to examine whether K76 or K120 are SUMO acceptor sites on DHX9. To validate DHX9 SUMOylation in cells, we expressed SFB (S-protein/

FLAG/Streptavidin-binding peptide)-tagged DHX9 alone or along with HA-SUMO1 or HA-SUMO2 in HeLa cells. Immunoprecipitation (IP) of SFB-DHX9 under denaturing conditions revealed a substantial conjugation with HA-SUMO2 compared to HA-SUMO1 (Fig. 1c). To further validate this, we used recombinant His-GST-DHX9$^{1-399}$ for in vitro SUMOylation assays, finding that His-GST-DHX9$^{1-399}$ was more significantly modified by SUMO2 compared to SUMO1 (Fig. 1d and Supplementary Fig. 1d). As the SUMOylation of several RNA helicases can be profoundly enhanced after stress[44], and DHX9 has a critical role in the DDR[26,45], we next explored the influence of DNA damage on DHX9 SUMOylation. Cells co-expressing SFB-DHX9 and HA-SUMO2 were treated with vehicle (DMSO), DNA replication inhibitor hydroxyurea (HU), and Topoisomerase I inhibitor camptothecin (CPT). CPT triggers single-strand breaks, which can transform into DSBs, causing fork collapse, while short-term HU exposure leads to replication stalling. DHX9 modification by HA-SUMO2 was profound without DNA damage and remained unchanged under DNA damage conditions (Fig. 1e). To ascertain if endogenous DHX9 is modified by SUMO2/3, DHX9 was immunoprecipitated from HeLa cells. The result showed that endogenous DHX9 was SUMO2/3 conjugated (Supplementary Fig. 1c).

To determine which lysine residues on DHX9 act as SUMO conjugation sites, we generated a series of DHX9 mutants, substituting K76, K120, and both K76/120 with arginines. Compared to wild-type DHX9 (DHX9$^{WT}$), the SUMOylation of DHX9 K120R mutant (DHX9$^{K120R}$) was markedly reduced, whereas K76R mutant (DHX9$^{K76R}$) remained unaffected (Fig. 1f). To confirm that K120 is the acceptor site for SUMO2/3, we expressed WT and different DHX9 mutants in HeLa cells. As observed previously, endogenous SUMO2/3 conjugates on SFB-DHX9$^{K120R}$ were substantially reduced to a level comparable to SFB-DHX9$^{2KR}$ (Fig. 1g), suggesting that K76 does not contribute to DHX9 SUMOylation. Consistent with the SUMOylation of DHX9 at K120 in cells, recombinant DHX9$^{1-399}$ fragments containing WT or K120R were tested for SUMOylation in vitro. The results showed again that DHX9$^{1-399}$ was preferably conjugated by SUMO2 rather than SUMO1 at K120, as introducing the K120R mutation to this DHX9 fragment markedly reduced the SUMOylation in vitro. (Fig. 1h and Supplementary Fig. 1e). We next created HeLa derivative clones harboring either the empty vector or expressing SFB-DHX9$^{WT}$ (WT-2) or SFB-DHX9$^{K120R}$ (K120R-13) controlled by a Tet-On system. The association of DHX9 and SUMO2/3 within the nucleus was assessed by proximity-ligation assay (PLA). All DHX9 variants were engineered to resist small interfering RNA (siRNA) targeting without altering their protein sequences. To avoid potential interference, endogenous DHX9 was silenced by siRNA, followed by doxycycline-induced expression of DHX9$^{WT}$ or DHX9$^{K120R}$. Fluorescent foci, representing the close association between SFB-DHX9 and SUMO2/3, were markedly detected in the nucleus of cells expressing SFB-DXH9$^{WT}$ but not SFB-DHX9$^{K120R}$ (Fig. 1i–k). These findings strongly support K120 within the predicted LKAE consensus SUMOylation motif as a major SUMOylation site of DHX9.

### DHX9 SUMOylation prevents DNA damage-associated cell death
When comparing the human DHX9 protein sequence across different species, we noted that K120 located in a consensus SUMOylation motif is highly conserved in most vertebrates (Supplementary Fig. 2a), suggesting the importance of SUMOylation at K120. To elucidate the biological role of K120 SUMOylation, we examined the impact of DHX9$^{K120R}$ on cell survival. HeLa cells transfected with control or DHX9 siRNA were introduced with the corresponding empty vector, SFB-DHX9$^{WT}$, or SFB-DHX9$^{K120R}$. Upon doxycycline induction for SFB-DHX9 expression, daily cell counts were recorded. Depletion of DHX9 by siRNA, cells expressing DHX9$^{K120R}$ exhibited reduced cell numbers similar to those carrying the corresponding empty vector. This trend was consistent in another cancer cell line, U2OS, underscoring the critical role of DHX9 SUMOylation in cell

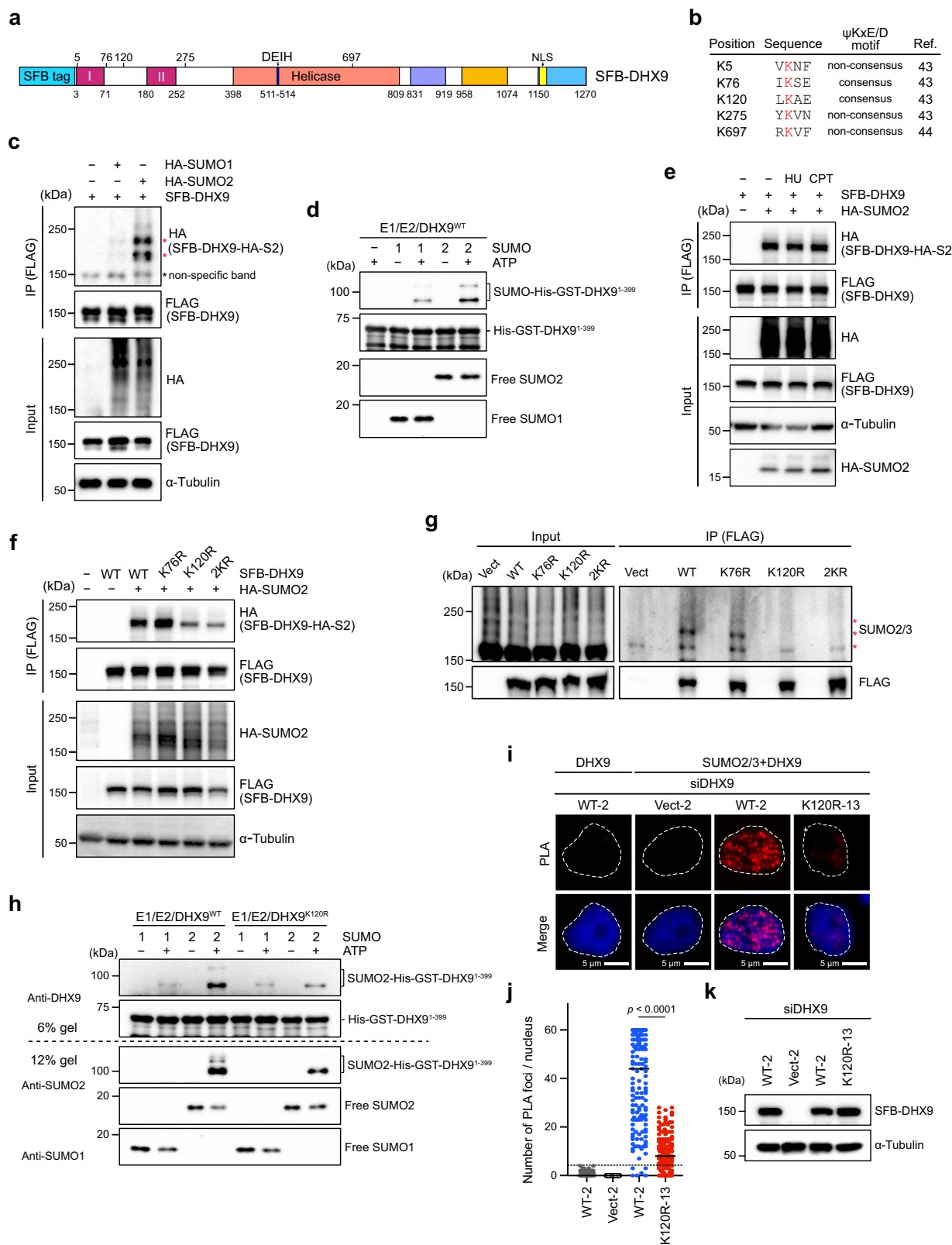

survive (Fig. 2a, b). Further analysis using Western blot validated that the reduced cell numbers in DHX9[K120R] were not due to insufficient protein expression (Fig. 2c). Colony formation assay further revealed that cells expressing DHX9[K120R] failed to restore colony numbers comparable to DHX9[WT] (Supplementary Fig. 2b). Using HeLa inducible clones, we observed similar cell proliferation defects in clones expressing either the corresponding empty vector or the DHX9[K120R],

transfected with siDHX9 (Fig. 2d). To ascertain whether the reduced cell numbers in DHX9[K120R] cells were due to increased cell death, we used Annexin V-APC assay to analyze cell death in different HeLa clones. In contrast to DHX9[WT], clones expressing DHX9[K120R] displayed a significant increase in cell death despite comparable protein levels between DHX9[K120R] and DHX9[WT] (Fig. 2e, f and Supplementary Fig. 2c).

**Fig. 1 | K120 is a major SUMO conjugation site of DHX9. a** A primary structure of DHX9, with an SFB-tag at its N-terminus, shows the helicase core and the location of five putative SUMO2 conjugation sites. Colored boxes represent specific features: the DExH-box colored in blue, the NLS in yellow, and boxes in magenta (I and II) for dsRBD1 and dsRBD2, respectively. **b** Sequences of five putative SUMO2 acceptor sites mapped by system-wide studies. **c** HeLa cells transfected with SFB-DHX9 (8 μg) alone or with HA-SUMO1 (4 μg) or HA-SUMO2 (3 μg) were used for FLAG affinity purification under denaturing conditions. Protein levels and SUMO conjugation on SFB-DHX9 were determined by Western blot with the indicated antibodies. **d** A His-GST-tagged DHX9 fragment (1–399) purified from *E. coli* was subjected to reconstituted reactions to examine DHX9 SUMOylation in vitro. Levels of recombinant DHX9$^{1-399}$ and SUMO-modified DHX9$^{1-399}$ were determined by Western blot using anti-DHX9 antibody. **e** DHX9 SUMOylation is not regulated by DNA damage. HeLa cells transfected with SFB-DHX9 alone or with HA-SUMO2 were treated with DMSO, HU (4 mM), or CPT (2 μM) for 1 h. SUMOylation of DHX9 and

protein levels were determined by Western blot. **f, g** DHX9 K120 is the major SUMO2 acceptor site. The HA-SUMO2 (**f**) and endogenous SUMO2/3 (**g**) conjugation of SFB-DHX9 variants purified by the anti-FLAG magnetic beads was analyzed by Western blot. **h** Bacterially expressed recombinant DHX9$^{1-399}$ (WT or K120R) were subjected to test DHX9 SUMOylation in vitro. DHX9 SUMOylation was assessed by Western blot. **i–k** SFB-DHX9$^{WT}$, but not SFB-DHX9$^{K120R}$ associated SUMO2/3 in the nucleus. HeLa derivative clones transfected with siDHX9 were used for expressing the vector, wild-type (WT), or K120R DHX9 upon doxycycline induction, followed by PLA analysis using the indicated antibodies. **i** Representative images of PLA analysis. **j** Quantification of PLA foci-positive cells (red). The black line indicates the median. The number of PLA foci per nucleus was counted from 150 nuclei under each condition and analyzed by a two-sided Mann–Whitney *U* test. **k** Levels of SFB-DHX9 in different stable cell lines were determined by Western blot. Source data are provided as a Source Data file.

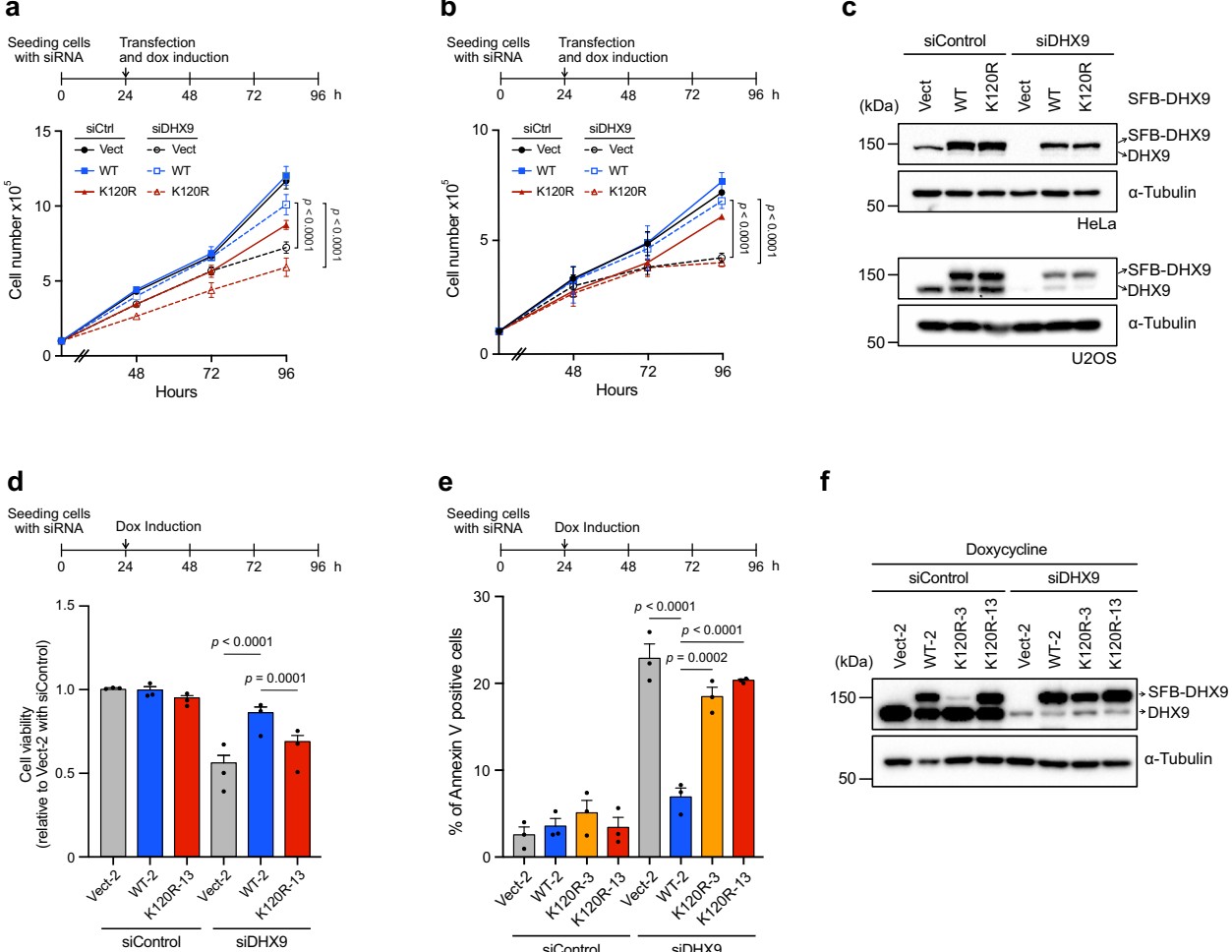

**Fig. 2 | DHX9 SUMOylation is crucial for cell survival. a, b** DHX9 SUMOylation supports cell proliferation in different cell types. As shown in the upper timeline, $1 \times 10^5$ of HeLa (**a**) or U2OS (**b**) cells were reverse transfected with siControl or siDHX9. 1 d later, the cells were forward transfected with the empty vector, or SFB-DHX9 variants, with doxycycline simultaneously added to the medium. Cell counts were then recorded daily in triplicates. Data are presented as mean ± SEM from $n \geq 3$ individual experiments analyzed by two-way ANOVA. **c** Protein levels of SFB-DHX9$^{WT}$ and SFB-DHX9$^{K120R}$ in cells used for cell counts in (**a, b**) were determined by Western blot. **d** HeLa derivative clones (Vect-2, WT-2, and K120R-13) transfected with siDHX9 were induced with doxycycline to express SFB-DHX9 variants. Four

days after ablating endogenous DHX9, cell counts were quantified. Data are presented as mean ± SEM with dots indicating the results of three independent experiments analyzed by one-way ANOVA. **e** Same as the condition in (**d**), cell death of indicated HeLa derivative clones was assessed using Annexin V apoptosis detection kits. Data are presented as mean ± SEM with dots indicating the results of three independent experiments analyzed by one-way ANOVA. **f** Protein levels of SFB-DHX9 variants in different HeLa derivative clones transfected with control or DHX9 siRNAs were determined by Western blot. Source data are provided as a Source Data file.

We next speculated whether cells with DHX9[K120R] display increased DNA damage after DHX9 knockdown. HeLa derivative clones transfected with control or DHX9 siRNAs for 4 days were treated with DMSO or CPT, and the DNA damage marker γH2AX was determined using immunofluorescence staining. In siDHX9 but not siControl transfected cells, we observed a marked increase in γH2AX foci-positive cells and elevated γH2AX foci intensity in clones expressing empty vector or DHX9[K120R] in the absence of exogenous threats (Fig. 3a–c). With CPT, all siControl transfected HeLa derivative clones displayed similar levels of CPT-induced γH2AX foci (Fig. 3b, c). Conversely, HeLa clones depleted of DHX9 for 4 days were largely lost during immunofluorescence staining following CPT treatment. As a result, the influence of DHX9[K120R] on CPT-induced γH2AX foci formation under this condition was not examined. To further confirm the correlation between increased γH2AX intensity and the expression of DHX9[K120R], we plotted the intensities of γH2AX and FLAG staining in a 2D format (Fig. 3d, e). Under DMSO treatment, most WT-2 cells were in the bottom right area, whereas K120R-13 cells were notably shifted to the top right area, correlating higher γH2AX intensity with DHX9[K120R] expression. The observations were not attributed to the failure of nuclear localization of DHX9[K120R], nor to different expression levels of DHX9[WT] or DHX9[K120R] in HeLa inducible lines (Fig. 3a, f). Consistent results were also observed in replicated experiments with transiently transfected SFB-DHX9 variants in DHX9-depleted HeLa cells (Supplementary Fig. 3a–f). These results support the crucial role of DHX9 SUMOylation in mitigating DNA damage and cell death.

## DHX9 SUMOylation suppresses R-loop accumulation and the impairment of transcription termination

One of the key functions of DHX9 is to resolve R-loop accumulation, which is a major threat to genome stability. Based on the observations that DHX9[K120R] cells displayed accumulated DNA damage and cell death, we hypothesized that DHX9[K120R] fails to suppress R-loop accumulation when endogenous DHX9 is depleted. We employed slot blot assays to assess global R-loop levels in HeLa cells, in which endogenous DHX9 was depleted, and SFB-DHX9[WT], SFB-DHX9[K120R], or the empty vector were transiently introduced (Fig. 4a). DNA/RNA hybrids extracted from cell lysates were either pre-treated with RNH or left untreated, then transferred onto nylon membranes and assessed with the S9.6 antibody, known for its high affinity to DNA/RNA hybrids[26,46]. Unlike SFB-DHX9[WT], cells carrying either the corresponding empty vector or SFB-DHX9[K120R] failed to prevent R-loop accumulation. The control group, in which cell extracts were pre-treated with RNH, showed no S9.6 signal, confirming that the signal detected in the RNH-minus group originated from R-loops. The same assay was repeated using HeLa derivative clones. For a duplicated set of HeLa clones, RNH-mCherry was transiently expressed before extracting DNA/RNA hybrids for slot blot assays (Supplementary Fig. 4a). Importantly, the increased R-loops in HeLa derivatives carrying the corresponding empty vector or SFB-DHX9[K120R] were significantly reduced upon transient RNH expression. Overall, the results highlight the importance of DHX9 SUMOylation in facilitating R-loop resolution.

DHX9 is enriched at the promoter-proximal regions of β-actin and γ-actin genes, and R-loop profiles of these genes in the absence of DHX9 have been documented in a previous study[26]. We next examined whether DHX9[K120R] impairs R-loop resolution at β-actin and γ-actin genes. For this purpose, we employed a previously established approach using the S9.6 antibody for DNA/RNA hybrids immunoprecipitation (DRIP) (Fig. 4b)[26]. HeLa cells with control or DHX9 siRNAs were transfected with the corresponding empty vector, SFB-DHX9[WT], or SFB-DHX9[K120R]. The R-loop profiles of β-actin and γ-actin genes were assessed using the DRIP analysis. We observed a significant increase of R-loops over the termination regions of the β-actin gene (5′ pause, pause, C, and D) and γ-actin gene (A, B, and C) in cells carrying the empty vector or SFB-DHX9[K120R]. In contrast, such an increase was not observed in cells with control siRNA or those carrying SFB-DHX9[WT] (Fig. 4c, d and Supplementary Fig. 4b). Notably, the enrichment of DRIP signals over the examined regions was sensitive to RNH treatment, validating the accumulation of R-loops.

The role of DHX9 in R-loop metabolism is crucial for transcription termination[26]. We next asked if DHX9 SUMOylation is involved in resolving R-loops in the polyA-proximal regions of β-actin and γ-actin genes. With RT-qPCR analysis, we found that cells lacking DHX9 and carrying the corresponding empty vector exhibited increased read-through transcripts downstream of the polyA signal (Fig. 4e, f and Supplementary Fig. 4c). Interestingly, cells expressing SFB-DHX9[WT] effectively reduced the levels of readthrough transcripts, while cells expressing SFB-DHX9[K120R] mirrored the transcription termination deficiency seen in empty vector-carrying cells. Furthermore, to validate our findings, we conducted additional experiments using HeLa inducible clones, and the results were similar to our initial observations (Supplementary Fig. 4d, e). The comparable level of SFB-DHX9 variants in tested HeLa clones verified that transcription termination defects did not result from the insufficient expression of SFB-DHX9[K120R] (Supplementary Fig. 4f). The findings suggest a role for DHX9 SUMOylation in curbing R-loop formation amid transcription termination.

## The vulnerability of DHX9[K120R] cells to genotoxic stress can be rescued by RNH

Given the critical role of DHX9 SUMOylation in suppressing R-loops and the increased DNA damage in cells expressing DHX9[K120R], we postulated that cancer cells with DHX9[K120R] might be more susceptible to genotoxic stress. HeLa derivative clones transfected with siDHX9 were induced to express either DHX9[WT] or DHX9[K120R], and treated with various genotoxic agents, including CPT, HU, cisplatin, and berzosertib, an ATR kinase inhibitor (Fig. 5a–d). Without induction, both WT and K120R clones showed comparable sensitivity to CPT and HU treatments, but upon doxycycline induction, DHX9[WT] cells showed improved viability while DHX9[K120R] cells remained sensitive. In addition to CPT and HU, vulnerability to cisplatin and berzosertib was also assessed. DHX9[K120R] cells exhibited increased susceptibility to these agents, a sensitivity not due to lower expression of DHX9[K120R] as confirmed by Western blot (Supplementary Fig. 5). To verify whether the accumulation of R-loop-associated DNA damage is the primary driver of vulnerability to genotoxic drugs, we examined the effect of preventing R-loop accumulation by RNH overexpression. We found that K120R clones with DHX9 depletion showed an increase in γH2AX-positive cells compared to the WT clone. Notably, RNH overexpression in all DHX9-depleted HeLa clones significantly reduced γH2AX-positive cells (Fig. 5e, f). Simultaneously, cell proliferation was assessed under the same conditions. While HeLa derivative clones with control siRNA displayed similar viable cell counts, clones harboring the empty vector or DHX9[K120R] showed a significant reduction (Fig. 5g). Importantly, overexpression of RNH in DHX9-depleted HeLa clones effectively restored cell numbers in clones carrying the empty vector or DHX9[K120R]. Overall, these findings indicate that RNH expression can mitigate the R-loop-associated defects caused by DHX9[K120R].

## DHX9[K120R] fails to efficiently associate with R-loops and R-loop interacting factors

To elucidate why DHX9[K120R] fails to suppress R-loop accumulation, we tested whether SUMOylation affects the helicase activity of DHX9. We immunoprecipitated SFB-DHX9 variants from HeLa cells, where WT DHX9 was further subjected to in vitro SUMOylation. The purified SFB-DHX9[WT] and SFB-DHX9[K120R] were then incubated with the DNA/RNA hybrids with a 3 prime RNA overhang for unwinding assay (Supplementary Fig. 6a). No significant unwinding difference was observed between SFB-DHX9[WT] and SFB-DHX9[K120R], suggesting that SUMOylation at K120 may not directly affect the helicase activity. We next asked

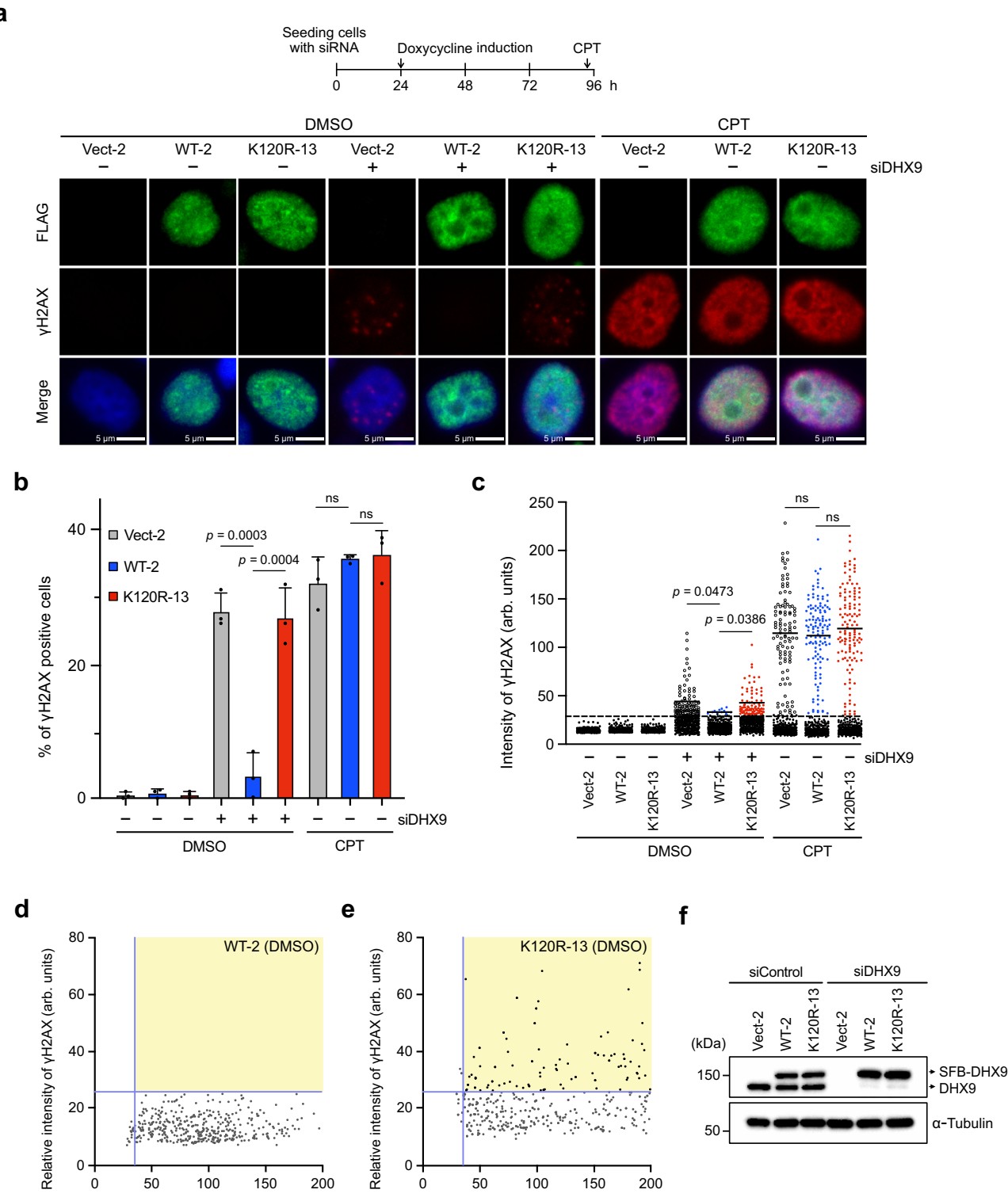

**Fig. 3 | DHX9 SUMOylation prevents the accumulation of DNA damage.**
**a**–**e** HeLa derivative clones transfected with negative control or DHX9 siRNAs for 1 d were induced to express SFB-DHX9$^{WT}$ (WT-2) or SFB-DHX9$^{K120R}$ (K120R-13) by doxycycline for 3 d, followed by treatment with DMSO or CPT (1 μM) for 1 h. The percentage and intensity of γH2AX (red) and FLAG (green) staining were determined. **a** Representative images of three independent experiments. Scale bar = 5 μm. **b** The percentage of γH2AX foci-positive cells was quantified. Data are presented as mean ± SD with dots indicating results from three independent experiments (ns = no significance) analyzed by one-way ANOVA. **c** Intensity of γH2AX foci was quantified in DMSO and CPT-treated HeLa clones. Representative data of three biological replicates analyzed by two-sided *t*-test. Cells with background staining were colored black. The black line indicates the median. **d**, **e** The correlation between SFB-DHX9 expression and γH2AX foci intensity in WT and mutant clone cells treated with DMSO were plotted in 2D, respectively. **f** Representative Western blot of three independent experiments shows the levels of endogenous and SFB-tagged DHX9 in different HeLa derivative lines. Source data are provided as a Source Data file.

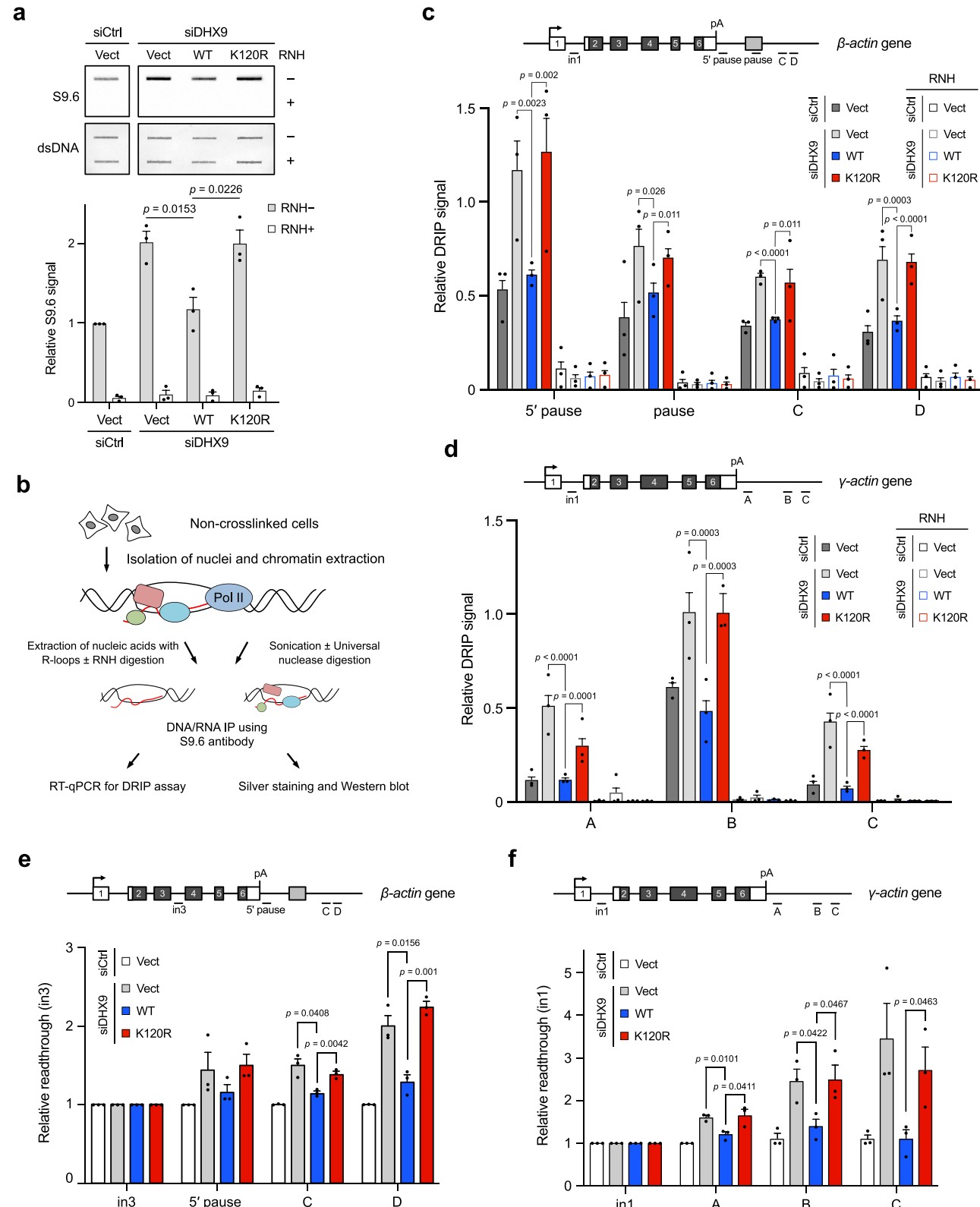

if DHX9^K120R fails to associate with R-loops using PLA analysis in DHX9-depleted cells (Fig. 6a). PLA analysis showed a robust association between SFB-DHX9^WT and R-loops. Conversely, PLA foci were significantly reduced in SFB-DHX9^K120R cells, denoting the reduced interaction between SFB-DHX9^K120R and R-loops. To ascertain this observation, we further immunoprecipitated DNA/RNA hybrids from the nuclear extracts of cells expressing SFB-DHX9 variants or the

empty vector using the S9.6 antibody to assess the binding of SFB-DHX9 variants with R-loops (Figs. 4b and 6b and Supplementary Fig. 6b). Unlike SFB-DHX9^WT, SFB-DHX9^K120R showed poor association with R-loops, suggesting the importance of DHX9 SUMOylation for its accumulation on R-loops.

Many studies have shown R-loop accumulation when RNA-processing factors are lacking, but the cooperative role of these

**Fig. 4 | DHX9 SUMOylation suppresses R-loop accumulation and the impairment of transcription termination. a** HeLa cells expressing DHX9$^{K120R}$ displayed an accumulation of R-loops. HeLa cells with DHX9 depletion by siRNA were transfected with the empty vector, SFB-DHX9$^{WT}$, or SFB-DHX9$^{K120R}$. DNA/RNA hybrids isolated from nuclear extracts were treated with or without RNH and analyzed using a slot blot assay with the S9.6 antibody. The relative S9.6 signal was quantified (mean ± SEM of three independent experiments). Top: representative blots; Bottom: quantification of relative S9.6 signal from $n = 3$ biological replicates analyzed by two-sided $t$ test. **b** Workflow of nucleic acids extraction and the RNH digestion, followed by DRIP assay. **c, d** HeLa cells treated with control or DHX9 siRNA were transfected with SFB-DHX9$^{WT}$, SFB-DHX9$^{K120R}$, or the empty vector. DRIP using the S9.6 antibody was analyzed by RT-qPCR for the abundance of R-loops over the polyA-proximal regions of $\beta$-actin and $\gamma$-actin genes, respectively. Values are normalized to intron 1 (in1) (mean ± SEM of three independent experiments analyzed by two-sided $t$ test. **e, f** Ablating K120 SUMOylation resulted in transcription termination deficiency. HeLa cells treated with control or DHX9 siRNA were transfected with SFB-DHX9$^{WT}$, SFB-DHX9$^{K120R}$, or the empty vector. Extracted nucleic acids were analyzed by RT-qPCR for readthrough transcription of $\beta$-actin and $\gamma$-actin genes, respectively. Data of three biological individual experiments. Values are normalized to β-actin intron 3 (in3) and γ-actin in1 and presented as mean ± SEM (two-sided $t$ test). Source data are provided as a Source Data file.

---

factors in R-loop resolution is not fully understood[23–31]. While a reduced association between DHX9$^{K120R}$ and R-loops was observed, it was not entirely abolished. One of the major functions of protein SUMOylation is to enhance protein-protein interactions. We next inquired whether DHX9$^{K120R}$ exhibits impaired interaction with the proteins involved in R-loop resolution. To explore this, we used mass spectrometry analysis to identify nuclear proteins interacting with endogenous DHX9. Among the identified proteins, we selected 345 with a minimum of 2 peptides detected through mass spectrometry for further functional group analysis using Ingenuity Pathway Analysis (Supplementary Fig. 6c). The functions of proteins co-precipitated with DHX9 are mainly linked to RNA metabolism and gene expression. By comparing our DHX9 interactome with previously published DNA/RNA hybrid interactome studies[26,47], the key hits involved in R-loop resolution were summarized (Supplementary Fig. 6d).

To validate whether DHX9 SUMOylation bolsters the interplay between DHX9 and its identified interactors, we pulled down SFB-tagged proteins extracted from cells expressing the corresponding empty vector or SFB-DHX9 variants. The results revealed that RNA interacting proteins like PARP1, RNA Pol II, XRN2, DDX21, PRP19, and SF3B1 were abundant in SFB-DHX9$^{WT}$ pull-down, but noticeably reduced in SFB-DHX9$^{K120R}$ pull-down (Fig. 6c, d). Furthermore, we conducted a parallel assessment using K76R, a mutation also located in the dsRBD1 and dsRBD2 linker region. In contrast to SFB-DHX9$^{K120R}$, the K76R mutation had no noticeable impact on DHX9's interaction with the tested proteins. Moreover, as a component of Pol II holoenzyme, it is noteworthy that DHX9$^{K120R}$ displayed a profound deficiency in binding Pol II, suggesting a crucial role of DHX9 SUMOylation in Pol II association. Besides, we immunoprecipitated endogenous XRN2, SF3B1, or DDX21 from cells expressing the empty vector or SFB-fusions. Western blot analysis revealed robust interactions of immunoprecipitated proteins with SFB-DHX9$^{WT}$ and SFB-DHX9$^{K76R}$, but not with SFB-DHX9$^{K120R}$ (Supplementary Fig. 6e–g). To directly link the observed effects to DHX9 SUMOylation, we mutated A121 or E122 in the LKAE SUMOylation motif, leaving K120 unchanged, and repeated the earlier experiments. The results showed that DHX9$^{A121G}$ acts similarly to DHX9$^{WT}$ in protein-protein interactions and R-loop associations. Conversely, DHX9$^{E122A}$, mirrored DHX9$^{K120R}$, exhibited binding defects with R-loops and numerous R-loop-binding proteins (Fig. 6e, f). Comparison of the SUMOylation of these DHX9 mutants showed that DHX9$^{A121G}$ remained SUMOylated by HA-SUMO2, while DHX9$^{E122A}$ exhibited reduced SUMO2 conjugation (Fig. 6g). PLA analyses further confirmed robust interactions of SFB-DHX9$^{WT}$, but not SFB-DHX9$^{K120R}$, with tested R-loop binding proteins (Supplementary Fig. 6h–k). The findings emphasize that DHX9 SUMOylation enhances its interactions with R-loops and associated proteins.

We next explored if SFB-DHX9$^{WT}$ interacts with R-loop-binding proteins dependent on RNA (Fig. 6h). Pre-treating nuclear extracts with RNase A prior to SFB-tag pull-down resulted in distinct protein association patterns with SFB-DHX9$^{WT}$. The association with RNA binding protein NONO was abolished, while interactions with SF3B1, ADAR1, and XRN2 were partially reduced by RNase A. Intriguingly, RNase A treatment did not impede the interactions of SFB-DHX9$^{WT}$

with Pol II, PARP1, and DDX21. The findings were further validated using PLA analysis, in which PFA-fixed cells with RNase A treatment did not abolish interactions of DHX9 with both PARP1 and DDX21, aligning with the SFB tag pull-down results (Supplementary Fig. 6l, m). To examine whether the interactions of DHX9 with PARP1 and DDX21 depend on DNA/RNA hybrids, HeLa cells were transfected with the empty vector, SFB-DHX9$^{WT}$, or SFB-DHX9$^{WT}$ plus WT or enzyme-dead RNH-mCherry constructs for IP (Fig. 6i). While RNH overexpression in cells is typically used to globally deplete DNA/RNA hybrids, only a slight reduction in the association of DHX9 with PARP1 or DDX21 was observed, with a minor increase in interactions seen in mutant RNH-transfected cells compared to those with SFB-DHX9 alone. Similarly, using RNH on fixed cells did not impair the interactions between DHX9 and PARP1 or DHX9 and DDX21. These findings suggest that DHX9 SUMOylation plays a critical role in binding R-loops and interacting with proteins in both RNA-dependent and RNA-independent manners.

## DHX9 SUMOylation enhances its association with PARP1 and DDX21 via SUMO-SIM interaction

The discovery that DHX9$^{WT}$, unlike DHX9$^{K120R}$, binds to various R-loop interacting proteins suggests that these interactions are SUMOylation dependent. Considering the known role of SUMO in promoting non-covalent bindings to proteins with SIMs, we hypothesized that R-loop binding proteins associated with SUMOylated DHX9 might possess SIMs. Specifically, we investigated SIMs in PARP1 and DDX21, which have been uncovered from studies of global non-covalent SUMO interaction networks[48,49]. While prior research identified a SIM in DDX21, SIMs in PARP1 have not been reported. To address this, we applied the GPS-SUMO program, predicting five SIM candidates in PARP1 (Fig. 7a and Supplementary Fig. 7a). Given the multiple predicted SIMs in PARP1, we next used AlphaFold-predicted 3D structure of PARP1 to pinpoint the most promising targets[50]. We focused on SIM1 and SIM2 in PARP1 because of their consensus SIM sequences and surface location on the PARP1 (Supplementary Fig. 7b). Upon analyzing the human PARP1 and DDX21 protein sequences across various species, it was observed that SIM1/2 in PARP1 and the only SIM in DDX21 exhibit high conservation among the compared vertebrates (Supplementary Fig. 7c). This conservation suggests a significant role for SIMs in both PARP1 and DDX21.

We next asked whether DHX9 SUMOylation enhances interactions with PARP1 and DDX21 through SIM recognition. In the HA affinity purification, we noticed that each single SIM mutant of PARP1, compared to HA-PARP1$^{WT}$, showed reduced interaction with endogenous DHX9 (Supplementary Fig. 7d). Based on this observation, we generated a double-SIM mutant (PARP1$^{sim1/2}$) to assess its interaction with DHX9. The co-IP assays revealed that HA-PARP1$^{sim1/2}$ markedly failed to associate with SFB-DHX9 in both FLAG and HA affinity purifications (Fig. 7b). Similar results were observed with HA-DDX21$^{sim}$, compared to HA-DDX21$^{WT}$, displaying a noticeable reduction in association with SFB-DHX9 (Fig. 7c, d). Building on these observations of SUMOylation dependency in DHX9 and its R-loop interacting proteins, we next depleted SUMO E2 UBC9 in HeLa cells to examine the effects of UBC9 on DHX9's interactions with previously identified interactors (Fig. 7e).

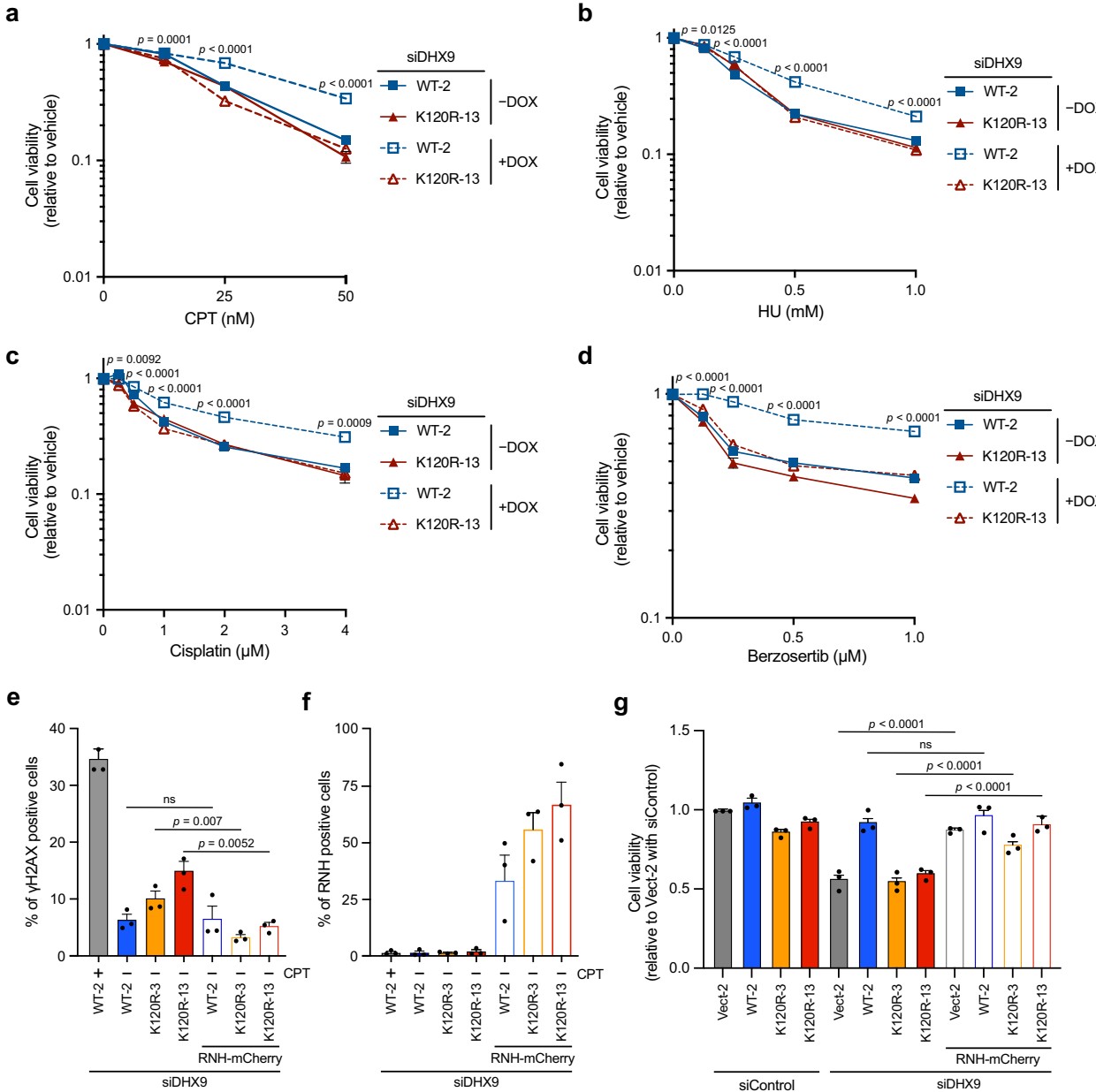

**Fig. 5 | The vulnerability of DHX9$^{K120R}$ cells to genotoxic stress can be rescued by RNH. a-d** DHX9$^{K120R}$ cells exhibited increased susceptibility to genotoxic drugs. HeLa inducible clones were transfected with siDHX9 to deplete endogenous DHX9. The expression of SFB-DHX9$^{WT}$ and SFB-DHX9$^{K120R}$ was induced with doxycycline. The cells were treated with CPT (**a**), HU (**b**), cisplatin (**c**), and berzosertib (**d**) at the indicated concentrations for 24 h. Cell viability was assessed using Cell Titer-Glo 2.0 and is presented as mean ± SEM of three independent experiments (two-way ANOVA). **e, f** Overexpression of RNH mitigated R-loop-associated DNA damage. HeLa inducible clones with DHX9 depletion were transfected with or without RNH.

The percentages of γH2AX-positive cells (**e**) and RNH-positive cells (**f**) were determined using Flow Cytometry. Data are presented as mean ± SEM of three independent experiments (ns = no significance) analyzed by two-sided *t*-test. **g** RNH overexpression rescued the cell survival of DHX9-depleted or DHX9$^{K120R}$ cells. HeLa inducible clones transfected with siDHX9 for 1 d were introduced with or without RNH for an additional 3 d before cell counting was performed. Data are presented as mean ± SEM of three independent experiments analyzed by two-sided *t*-test. Source data are provided as a Source Data file.

Unlike siControl transfected cells, where SFB-DHX9 stably associated with multiple RNA interacting proteins, the siUBC9 transfected sample showed remarkable reduction in associations between SFB-DHX9 and those proteins. These findings highlight the critical role of SUMO-SIM interactions in bridging DHX9 and multiple RNA interacting proteins.

**Fusion of a SUMO2 to DHX9$^{K120R}$ bypasses the SUMOylation of DHX9**

Attaching a SUMO or linear poly-SUMO to a SUMOylation-deficient protein is a common method for studying the roles of protein SUMOylation[51–53]. We next fused a SUMO2 to the N-terminus of DHX9$^{K120R}$ and examined whether SUMO2-DHX9$^{K120R}$ (S2-DHX9$^{K120R}$) compensates for deficiencies of DHX9$^{K120R}$. We compared the interactions of DHX9$^{K120R}$ and S2-DHX9$^{K120R}$ with PARP1 and DDX21 (Fig. 8a and Supplementary Fig. 8a), finding that S2-DHX9$^{K120R}$ exhibited increased co-precipitation with HA-PARP1 and HA-DDX21, but failed to interact with HA-PARP1$^{sim1/2}$, confirming that S2-DHX9$^{K120R}$ interacts with PARP1 through SUMO-SIM. These results demonstrate that a SUMO2 fusion on DHX9 enhances its associations with PARP1 and DDX21.

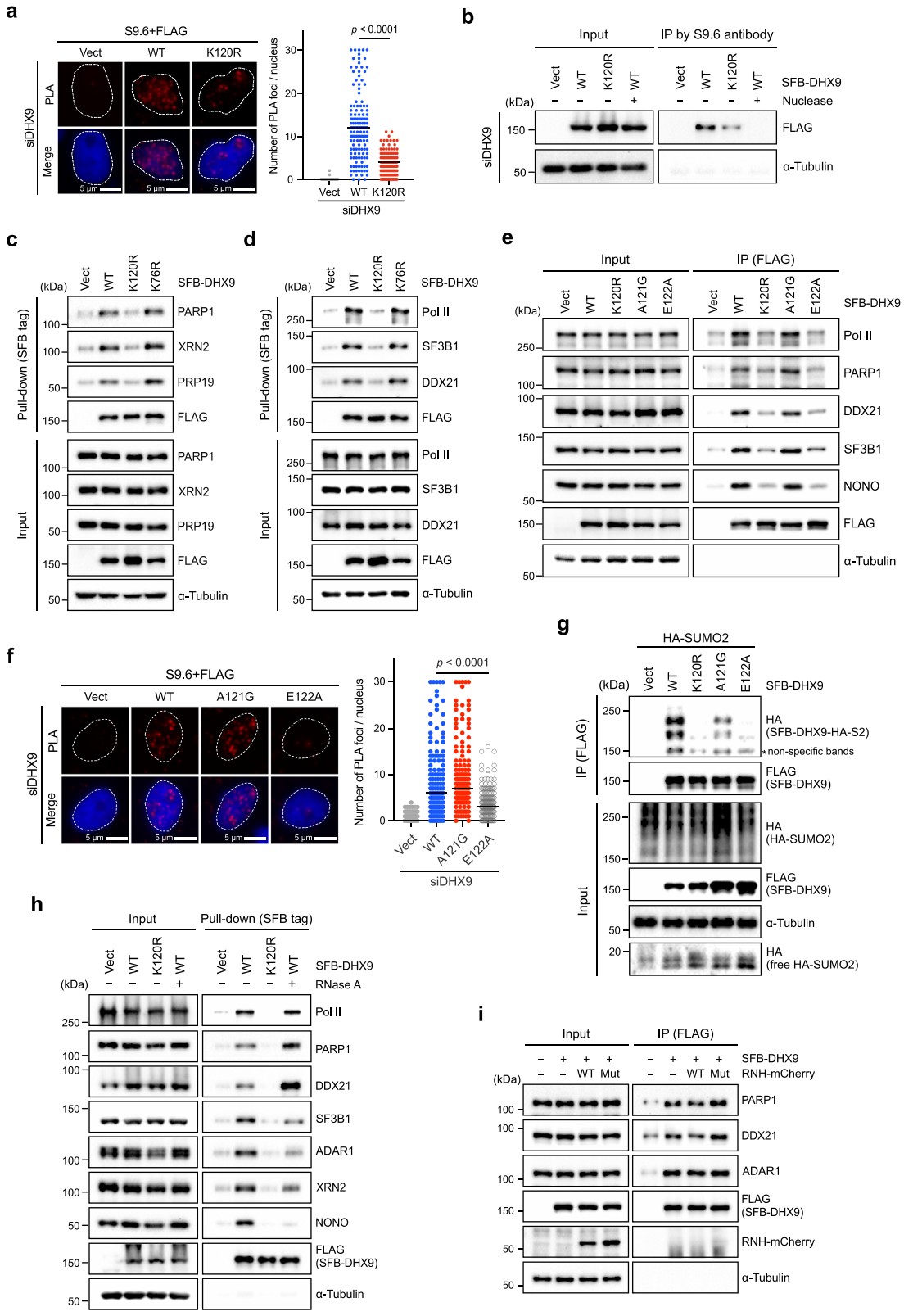

We then determined whether S2-DHX9$^{K120R}$ reduces the R-loop level in cells. PLA analysis showed that S2-DHX9$^{K120R}$, expressed at similar levels to other DHX9 variants, significantly increased association with R-loops compared to DHX9$^{K120R}$ (Fig. 8b and Supplementary Fig. 8b). Further evaluation at specific gene loci in cells demonstrated that S2-DHX9$^{K120R}$ reduces the R-loop level at examined genes (Fig. 8c and Supplementary Fig. 8c, d). Counting viable cells three days post-transfection with various DHX9 constructs in siDHX9 cells revealed a significant increase in cell viability with S2-DHX9$^{K120R}$ (Fig. 8d and Supplementary Fig. 8e). The fusion of SUMO2 to DHX9$^{K120R}$ restores R-loop binding, thereby reducing R-loop levels in cells and alleviating survival defects associated with DHX9$^{K120R}$. Together, our findings suggest that DHX9 SUMOylation is crucial for its interactions with RNA-processing proteins and for maintaining R-loop balance.

**Fig. 6 | DHX9$^{K120R}$ fails to efficiently associate with R-loops and R-loop interacting factors. a** The association of DHX9 variants with R-loops (S9.6 signal) in DHX9-depleted HeLa cells was assessed by PLA with indicated antibodies. The number of PLA foci (red) per nucleus was quantified by analyzing 150 nuclei under each condition (two-sided Mann–Whitney *U* test). The black line indicates the median. **b** SFB-DHX9$^{K120R}$ exhibited poor binding to R-loops. DRIP was prepared as described in Fig. 4b, with the addition of universal nuclease as a negative control. The association of SFB-DHX9 variants with R-loops was analyzed by Western blot, three biological repeats. **c, d** Ablating K120 SUMOylation attenuated the interaction between DHX9 and multiple RNA-processing factors. HeLa cell lysates were used for SFB-tag pull-down using Dynabead M-280-streptavidin. Proteins interacting with SFB-DHX9 variants were determined by Western blot, two biological repeats. **e** The associations between indicated SFB-DHX9 variants and multi-RNA binding proteins were determined by Western blot, three biological repeats. **f** Similar to (**a**), the

association of different SFB-DHX9 variants with R-loops in DHX9-depleted HeLa cells was assessed by PLA (red) with indicated antibodies (two-sided Mann–Whitney *U* test). **g** The SUMOylation of SFB-DHX9 variants purified by the anti-FLAG magnetic beads was analyzed by Western blot The association of SFB-DHX9 variants with R-loops was analyzed by Western blot, two biological repeats. **h** RNase A treatment showed distinct interaction patterns between DHX9 and RNA binding proteins. HeLa cell lysates treated without or with RNase A were used for SFB-tag pull-down. Proteins interacting with SFB-DHX9 variants were determined by Western blot, two biological repeats. **i** HeLa cells transfected with the empty vectors, SFB-DHX9, or SFB-DHX9 with WT or mutant RNH-mCherry plasmids were subjected to IP by anti-FLAG beads. The interaction between SFB-DXH9 and indicated proteins was determined by Western blot, two biological repeats. Source data are provided as a Source Data file.

## Discussion

In this study, we explore the pivotal role of DHX9 SUMOylation in regulating R-loop balance, a key factor in genome stability. While DHX9 is known to suppress R-loop accumulation during transcription-replication conflicts or genotoxic stress, the specific mechanisms of its regulation are not fully understood. We demonstrate that DHX9 is SUMOylated at K120 by SUMO2/3, essential for maintaining cell survival and genomic integrity. Lack of DHX9 SUMOylation at K120 disrupts R-loop balance, leading to DNA damage and cell death. Overexpressing RNH in DHX9 SUMOylation-deficient cells confirms the role of R-loops in these defects. Importantly, we show the crucial role of SUMO-SIM interactions in the functionality of DHX9 and multiple R-loop interacting proteins, highlighting the importance of DHX9 SUMOylation in managing aberrant R-loops through enhanced binding and interaction with R-loop-associated factors. While previous studies have shown DHX9 interacting with UBC9 for SUMO1 conjugation in its N-terminal region, our research indicates that in cells, DHX9 is primarily modified at K120 by SUMO2/3. Despite this, the K120R mutation in DHX9 doesn't eliminate all SUMO conjugation signals, hinting at additional modification sites. Emerging evidence has highlighted SUMOylation as a key regulatory factor in RNA metabolism, but its precise impact on RNA processing is not fully understood[39]. RNA helicases like DDX5, DDX19, and SETX, essential for specific functions in RNA processing, undergo SUMOylation, impacting their functions in RNA processing differently. For example, SUMOylation of DDX5 stabilizes its role in miRNA maturation[54], SUMOylation of DDX19 boosts mRNA export[55], and stress-enhanced SUMOylation of SETX promotes R-loop removal[41]. However, the functional implications of DHX9 SUMOylation remain unclear. Our study suggests that DHX9 SUMOylation at K120 is involved in R-loop resolution. Replacing K120 with arginine in DHX9 doesn't impact protein stability or nuclear localization but results in elevated cell death, accompanied by DNA damage and R-loop accumulation. Unlike SETX and other RNA processors affected by cellular stresses[44,56], DHX9 SUMOylation remains unaffected by CPT and HU, suggesting a distinct regulation mechanism. Whether other genotoxic agents or stresses have impacts on DHX9 SUMOylation remains to be elucidated.

Recent studies identify DHX9 as a critical regulator of R-loop levels[26,57–60], highlighting its role alongside the loss of other RNA-processing factors in complex R-loop regulation. How does DHX9 SUMOylation influence R-loop homeostasis? DHX9 and SETX specifically mitigate R-loops at transcription termination sites of *β*-actin and *γ*-actin genes, crucial for Pol II pause-dependent termination[26]. Furthermore, PARP1 senses R-loops and triggers the recruitment of DDX1, DDX5, and DDX18 to R-loop site[31,61]. Intriguingly, DHX9 collaborates with PARP1 against stress-induced R-loops, yet its chromatin association is independent of PARP1[26]. We show that DHX9 SUMOylation strengthens its interaction with multiple RNA-processing proteins, revealing distinct interaction patterns. While the associations between DHX9 and Pol II, PARP1, and DDX21 are RNA-independent, the

interactions between DHX9 and SF3B1, XRN2, and ADAR1 are partially RNA-dependent. Furthermore, RNH treatment does not abolish the associations of DHX9 with PARP1 and DDX21, suggesting another mechanism in bridging DHX9 and R-loop interacting proteins. Given that R-loops in mammals tend to be significantly longer, ~1 kb or greater[7], a coordinated effort of RNA-processing proteins is likely crucial for efficiently removing unscheduled R-loops. Recent studies have shown that RNA-dependent DEAD-box ATPases (DDXs) regulate RNA-containing phase-separated organelles, forming compartments that manage RNA-processing steps[62]. R-loop proteomics has identified hundreds of R-loop binding proteins with functional interactions in RNA metabolism, further highlighting the potential collaborative regulation of complex RNA-processing factors in R-loop balance[25,26]. Interestingly, proteins involved in RNA processing, including SETX, DDX3X, DHX15, DDX5, and DDX21, have been identified as top hits in global SUMO-binding studies[48,49], suggesting their potential associations with SUMOylated proteins in RNA processing. Similar to DDR proteins, where SUMO-SIM interactions enhance effective damage signaling and DNA repair[53,63], SUMOylation is vital for the interactions of DHX9 with a group of RNA-processing proteins. DHX9 SUMOylation recognizes the SIMs of PARP1 and DDX21, and without SUMOylation or with mutations in the SIMs of these proteins, the associations with DHX9 are compromised.

DHX9 unwinds DNA/RNA hybrids, dsRNA, and dsDNA, with a preference for RNA-containing duplexes over dsDNA[34]. The N-terminal region of human DHX9 and its *Drosophila* ortholog MLE, both capable of binding dsRNA[64–66], contains dsRBD1 and 2, connected by a linking loop (L1). Studies indicate that the dsRBD1-L1 region is not required for RNA binding and unwinding[66–68]. Moreover, K120 within the L1 linker of DHX9 is absent in MLE, suggesting that SUMOylation may not be essential for its helicase function. Our DNA/RNA hybrid unwinding assay further confirms that SUMOylation of DHX9 at K120 is dispensable for its helicase activity. Considering the position of K120 and the flexibility of the dsRBD1-L1 region, we speculate that SUMOylation at K120 offers an extra structural surface, enhancing DHX9's interactions with R-loop-associated proteins via SUMO-SIM interactions. Attaching a SUMO2 moiety to DHX9$^{K120R}$ restored its association with R-loops, potentially through increased interactions with PARP1, DDX21, and possibly other R-loop-binding proteins. The flexibility of the L1 loop likely allows the fused SUMO2 to move freely and able to mimic the native SUMOylation of DHX9.

Recently, our group discovered that ATR-phosphorylated DHX9 at S321 plays a crucial role in preventing R-loop accumulation following genotoxic stress[57]. Unlike DHX9 pS321, SUMOylation at K120 is not triggered by HU or CPT. Our findings suggest that SUMOylated DHX9, associated with RNA Pol II and other RNA-processing factors, plays a critical role in managing R-loop levels during transcription. Upon genotoxic stress, DHX9 pS321 facilitates its association at transcription-replication conflicts but still requires

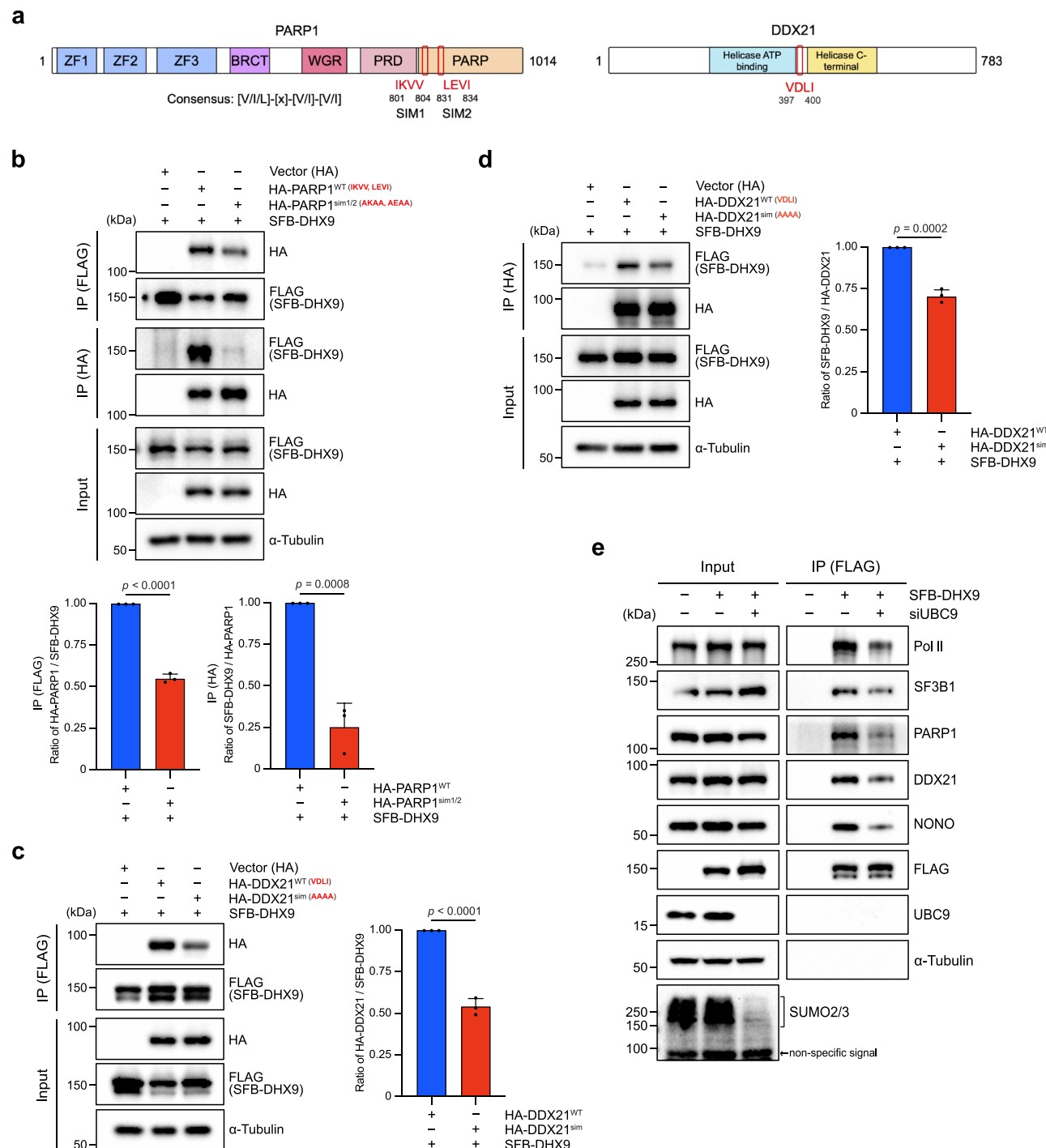

**Fig. 7 | DHX9 SUMOylation enhances its association with PARP1 and DDX21 via SUMO-SIM interaction. a** Schematic diagrams show predicted SIMs in PARP1, and a previously identified SIM in DDX21, with their locations highlighted in red boxes. **b** HeLa cells co-transfected with indicated constructs were split into two groups. One for FLAG affinity purification and the other for HA affinity purification. The proteins precipitated were determined by Western blot, and the quantification data are presented as mean ± SD from three separate experiments analyzed by two-sided *t* test. **c**, **d** HeLa cells co-transfected with indicated constructs were used for IP. One group of samples were subjected to FLAG affinity purification (**c**) and the other group of samples were subjected to HA affinity purification (**d**). The association of SFB-DHX9 with HA-DDX21 variants was determined by Western blot, and the quantification data are presented as mean ± SD from three separate experiments analyzed by a two-sided *t* test. **e** The SUMO pathway modulates protein associations between SFB-DHX9 and multiple RNA-processing factors. HeLa cells transfected with siControl or siUBC9 were transfected with indicated constructs, followed by FLAG affinity purification. Associations between DHX9 and RNA-processing factors were determined by Western blot (*n* = 3). Source data are provided as a Source Data file.

SUMOylation to link with R-loop resolving proteins. Investigating the potential interplay between phosphorylation and SUMOylation of DHX9 in regulating R-loops would be intriguing for future research. Our study suggests that DHX9 SUMOylation acts like a molecular glue, enhancing its interactions with RNA-processing factors crucial for RNA metabolism. Yet, the precise spatiotemporal coordination between SUMOylated DHX9 and these factors in R-loop dynamics awaits further elucidation.

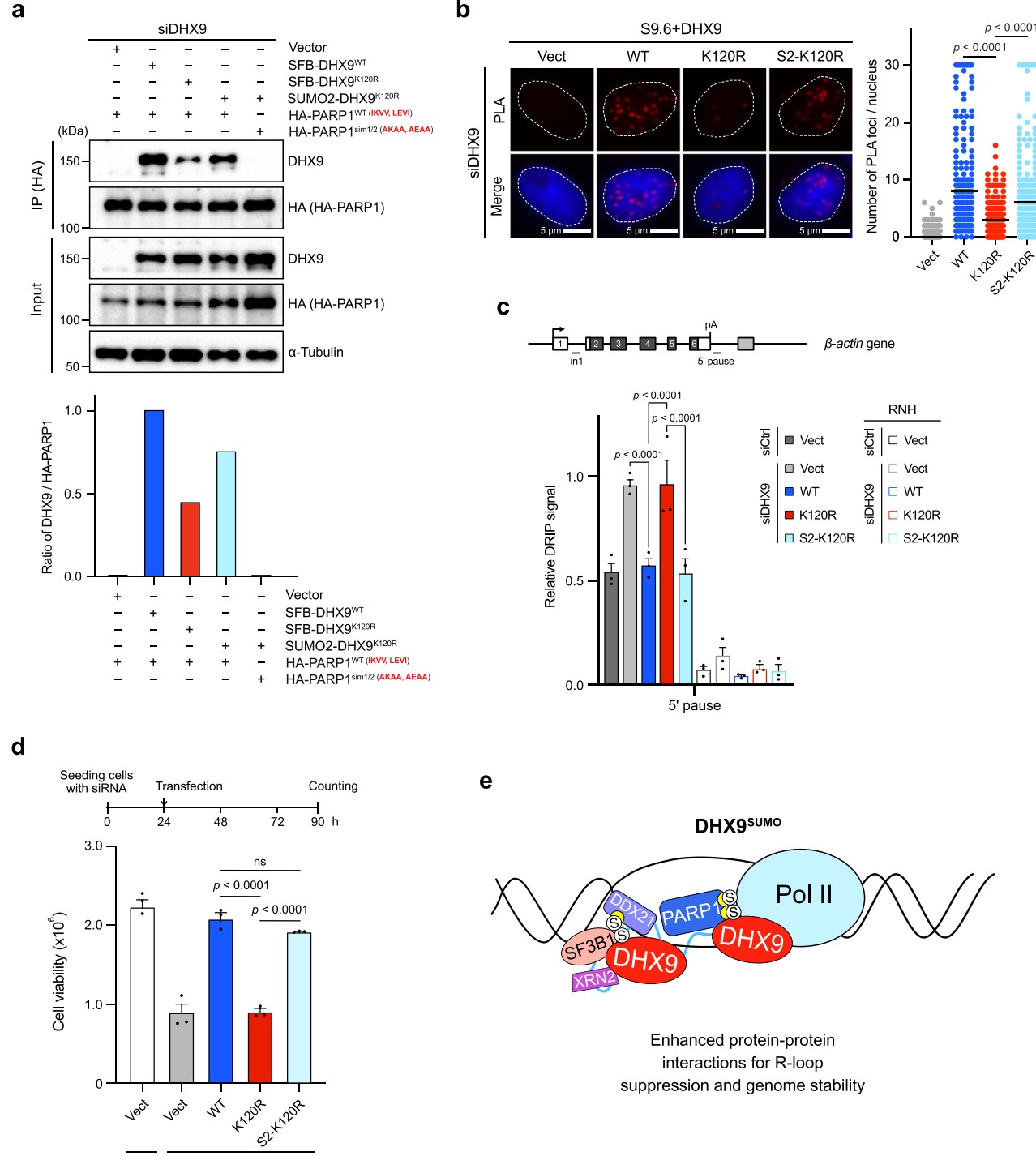

**Fig. 8 | Fusion of a SUMO2 to DHX9^K120R bypasses the SUMOylation of DHX9.**
**a** The interactions between different SFB-DHX9 variants and HA-PARP1 constructs were assessed by co-IP. HeLa cells co-transfected with indicated constructs were collected for IP using anti-HA beads. Levels of SFB-DHX9 variants and SUMO2-DHX9^K120R co-precipitated by HA-PARP1 variants were determined and quantified by Western blot. Representative data of two biological replicates. **b** The association between DHX9 variants and R-loops in DHX9-depleted HeLa cells was assessed by PLA with indicated antibodies. The number of PLA foci (red) per nucleus was quantified by analyzing 150 nuclei in each condition (two-sided Mann-Whitney $U$ test). The black line indicates the median. **c** The relative R-loop level at the 5′ pause region of $\beta$-actin gene was determined by RT-qPCR. The representative bar graph

was from three separate experiments. Values are normalized to $\beta$-actin in1 and presented as mean ± SEM. **d** S2-DHX9^K120R improved cell viability. HeLa cells with DHX9 depletion by siRNA were transfected with the indicated constructs for 66 h before collection. Cell counting for each condition in triplicates was obtained. Data are presented as mean ± SEM from three individual experiments analyzed by one-way ANOVA. **e** A model depicts the role of DHX9 SUMOylation in enhancing the association of DHX9 with R-loops and multiple RNA-processing factors, aiding in R-loop balance and genome stability. Yellow crescent shapes in PARP1 and DDX21 represent SIMs, and circled S symbols represent SUMO modifications. Source data are provided as a Source Data file.

## Methods

### Cell culture, transfection, and chemicals

HeLa and U2OS cells obtained from the ATCC were cultured in Dulbecco's modified Eagle's medium (DMEM, ThermoFisher) supplemented with L-glutamine and 10% fetal bovine serum (FBS, ThermoFisher) at 37 °C with 5% $CO_2$. Mycoplasma contamination was routinely monitored using MycoAlert PLUS Mycoplasma detection assay (LT07-710, Lonza). For RNAi transfection, 3 nM of Silencer Select siRNA was reverse transfected into cells using Lipofectamine RNAiMax (13778150, ThermoFisher) following the manufacturer's protocol. Details regarding the siRNA used in this study are summarized in Supplementary Table 1. For transient transfection of SFB-DHX9 variants, Lipofectamine 2000 (11668019, ThermoFisher) was used in the complementary experiments (Figs. 2a, b, 4, and Supplementary Fig. 2b) while polyethyleneimine (PEI) (Sigma-Aldrich) was used in the rest experiments. The small molecule inhibitors used to treat cells in this work were summarized in Supplementary Table 2.

### Plasmids and inducible clones

The coding sequence of DHX9 was cloned into a gateway cloning ENTRY vector using the pENTR/D-TOPO cloning kit (45-0218, ThermoFisher). For the siDHX9-resistant construct, silent mutations were introduced into the pENTRY-DHX9 vector without altering the DHX9 protein sequence[57]. The siDHX9-resistant vector was subsequently used for site-directed mutagenesis to substitute K120 with arginine using PCR and KLD enzyme mix reaction (M0554, NEB) following the manufacturer's protocol. To generate Tet-On-regulated SFB-DHX9 variants, coding sequences of SFB- DHX9 variants were lifted by PCR containing a NheI site at the N-terminus and an AgeI site at the C-terminus. The digested PCR fragment was ligated to the pTetOn3G vector. The RNH-mCherry plasmid is from Dr. Hsueh-Ping Catherine Chu's lab. To create inducible cell lines, HeLa cells were transfected with linearized pTetOn3G-SFB-DHX9 vectors or the corresponding empty vector followed by hygromycin (250 μg/mL, ThermoFisher) selection for up to 12 d. Each candidate clone was verified by Western blotting. To express the WT and SIM mutant HA-PARP1 and HA-DDX21, the coding DNA sequences of PARP1 and DDX21 were cloned into ENTRY vectors using pENTR/D-TOPO kit. The sim mutations were introduced to the corresponding ENTRY vectors by site-directed mutagenesis followed by the KLD enzyme mix reaction. Genes of interest in pENTRY plasmids were shuttled to destination vectors (pDEST-HA) using LR clonase II enzyme mix (11791020) for making HAx3-fused expression clones. To generate recombinant DHX9 proteins for in vitro SUMOylation assays, the pWH36 plasmid harboring the $His \times 8$-GST-TEV-DHX9$^{WT}$ was used for PCR-based deletion and the KLD enzyme mix reaction. Generated $His \times 8$-GST-TEV-DHX9$^{1-399}$ plasmid was further used to make the K120R variant by site-directed mutagenesis. To create SUMO2$^{12-91}$-fused DHX9$^{K120R}$, we removed the SFB-tag from SFB-DHX9$^{K120R}$, establishing an AgeI site to insert AgeI-SUMO2$^{12-91}$-AgeI. Primers used for plasmid construction are summarized in Supplementary Table 3.

### Antibodies

The following antibodies were used in Western blotting where indicated. SUMO1 (#4930, 1:1000), NONO (#90336, 1:1000), and UBC9 (#4786, 1:1000) were obtained from Cell Signaling Technology; PRP19 (sc-514338, 1:500), DHX9 (sc-137232, 1:500), DDX21 (sc-376953, 1:500), SF3B1 (sc-514655, 1:500), XRN2 (sc-365258, 1:400), Pol ll (sc-47701, 1:400), PARP1 (sc-8007, 1:400), PARP1 (sc-74470, 1:400), ADAR1 (sc-271854, 1:500), and DsRed (sc-390909, 1:500) antibodies were from Santa Cruz; DHX9 (ab26271, 1:2000) and SUMO2/3 (ab3742, 1:1000) were purchased from abcam; pRPA32 S33 (A300-246A, 1:2000) was from Bethyl Lab; FLAG (F7425, 1:2000) and α-Tubulin (T6074, 1:12000) were purchased from Merck; HA (715500, 1:2000), goat anti-mouse-

HRP (G21040, 1:10,000), and goat anti-rabbit-HRP (G21234, 1:10,000) were from ThermoFisher.

### IP, co-IP, SFB tag pull-down, and Western blotting

To determine the interaction between SFB-DHX9 variants and HA-tagged proteins or multiple R-loop interacting proteins in cells, cells were lysed in NETN buffer (20 mM Tris-HCl pH 8.0, 0.5 mM EDTA, 150 mM NaCl, 0.5% Igepal, 10% glycerol, 15 mM NEM) containing protease and phosphatase inhibitor cocktail (78440, ThermoFisher). Lysates were briefly sonicated, followed by centrifugation at $16,000 \times g$, 4 °C for 10 min to remove debris. 2% of each resulting supernatant was aliquoted as input, while the rest portion was used for IP (2 μg of anti-XRN2, SF3B1, or DDX21) at 4 °C overnight, SFB pull-down by the Dynabeads M-280 streptavidin at 4 °C for 4 h, or HA-tag IP with anti-HA beads (88837, ThermoFisher) at room temperature for 1 h. In certain cases, RNase A (20 μg/mL) or Universal Nuclease (88700, ThermoFisher) was concurrently incubated with lysates at room temperature for 30 min before switching to 4 °C overnight. For IP endogenous XRN2, SF3B1, and DDX21, 20 μl of protein G-conjugated magnetic beads (10004D, ThermoFisher) were added to the overnight IP samples for 1 h. Following IP or SFB-tag pull-down, the beads were washed 3 × 5 min with NETN buffer, and the captured proteins were lysed in 2X SDS sample buffer (100 mM Tris-HCl, 2% SDS, 200 mM DTT, 0.04 % Bromophenol blue, and 20% glycerol) for subsequent assessment by Western blot.

For Western blotting, cell lysates in 2X sample buffer underwent electrophoresis by SDS-PAGE and were transferred onto PVDF membranes (Immobilon-P, Millipore). Membranes were then blocked with 5% skim milk in TBS-T (1X TBS, 0.1% Tween-20) for 30 min, followed by incubation with primary antibodies for 2 h at room temperature or overnight at 4 °C. Post-incubation, the membranes were washed 3 × 8 min with TBS-T, followed by the incubation of HRP-conjugated secondary antibodies at room temperature for 1 h, 3 × 8 min wash with TBS-T. The signals were detected using chemiluminescence reagents (34580, ThermoFisher, and 170562, Bio-Rad) and imaged by the ChemiDoc imaging system (Bio-Rad).

### In vivo SUMOylation by denaturing IP

To test the SUMOylation of DHX9, we followed the previously published methods[53]. For the IP of SFB-DHX9, a 30 μl slurry of anti-FLAG-conjugated magnetic beads was mixed with the supernatants of denatured lysates and incubated at 4 °C overnight. For the IP of endogenous DHX9, the denatured lysates were mixed with 4 μg of either normal rabbit IgG (2729, Cell Signaling Technology) or anti-DHX9 (ab26271, abcam), 4 °C overnight. The next day, 30 μl of Dynabeads protein G (10004D, ThermoFisher) were added to the overnight mixture and incubated for 1 h. Following IP, the beads were washed three times with NETN buffer, and the captured proteins were lysed in 2X SDS sample buffer for subsequent assessment by Western blot.

### In vitro SUMOylation assay

*E. coli* BL21 RIL strain transformed with $His \times 8$-GST-TEV-DHX9$^{1-399}$ (WT or K120R) plasmids were cultured in LB broth at 37 °C. When $OD_{600}$ reached 0.4–0.6, 0.6 mM of isopropyl β-D-1-thiogalactopyranoside (IPTG) was added overnight at 24 °C to induce the expression of recombinant proteins. Recombinant DHX9 fragments were purified with cOmplete His-Tag Purification Resin (Roche) in GST buffer (50 mM Tris-HCl pH 8.0, 200 mM NaCl, 10% glycerol, 5 mM DTT, 0.1% triton X-100) at 4 °C for 1 h, followed by 4 washings in GST buffer containing 5 mM imidazole. Purified proteins were eluted in the GST buffer containing 400 mM imidazole. In vitro SUMOylation was performed using the SUMOylation kit (ab139470, abcam) with 1 μg of purified recombinant DHX9 at 37 °C for 1 h, followed by adding 6X SDS sample buffer.

## DHX9 interactome by mass spectrometry

Collected HeLa cells were resuspended in CSK buffer (10 mM PIPES pH 7.4, 100 mM NaCl, 3 mM MgCl$_2$, 300 mM sucrose) containing protease and phosphatase inhibitors. Cells were permeabilized with 0.1% Triton X-100 for 4 min on ice. After centrifugation at $1400 \times g$, 4 °C for 4 min, pellets were washed with CSK buffer and resuspended in NETN buffer containing protease and phosphatase inhibitors. Samples were then sonicated by Qsonica q700 and spun at $16000 \times g$, 4 °C for 10 min to remove debris. The resulting supernatants were pre-cleared with protein G beads, followed by IP using anti-DHX9 (sc-137232, Santa Cruz) or IgG (sc-2025, Santa Cruz) overnight at 4 °C. The next day Dynabeads protein G was added to the mixtures for 30 min capturing the DHX9-antibody complex and control IgG. Each sample ($n = 1$) was washed $4 \times 5$ min with NETN buffer and dissolved in 2X SDS sample buffer for SDS-PAGE electrophoresis and QC Colloidal Coomassie Blue staining (1610803, Bio-Rad). Protein gel was submitted to the Taplin Mass Spectrometry Facility, Harvard Medical School, Boston, USA, for further proteomic analysis. A detailed method of the mass spectrometry analysis was included in the Supplementary information.

## Immunofluorescence

HeLa cells or inducible clones on coverslips were permeabilized with 0.5% Triton X-100 in PBS for 5 min on ice, washed with PBS, fixed with 4% paraformaldehyde (PFA) in PBS at room temperature for 15 min, followed by PBS-0.5% Triton X-100 incubation for 3 min. Cells were incubated with a blocking solution (3% BSA and 0.05% Tween-20 in PBS) at room temperature for 30 min. The fixed cells were then incubated with anti-γH2AX antibody (05636, Merck, 1:250) diluted in blocking solution at 4 °C overnight, followed by anti-FLAG antibody (F7425, Merck, 1:500) at room temperature for 2 h. Following incubation, coverslips were washed $3 \times 5$ min with PBST (0.05% Tween-20 in PBS) and incubated with goat anti-mouse Alexa Fluor-594 (A-11005, ThermoFisher, 1:400) and goat anti-rabbit Alexa Fluor-488 (A-11034, ThermoFisher, 1:400) at room temperature in the dark for 1 h. Nuclei were counterstained with DAPI (D9542, Sigma-Aldrich) for 10 min, and coverslips were mounted using ProLong Gold Antifade Mountant (P36930, ThermoFisher), imaged with EVOS M7000 Imaging System. The intensity of FLAG and γH2AX staining were analyzed using Celleste (ThermoFisher).

## Proximity-ligation assay

In situ PLA analysis was carried out using the Duolink PLA kit (Merck). Cells on coverslips (ϕ 15 mm) were incubated with PBS-0.5% Triton X-100 on ice for 5 min, fixed with 4% PFA in PBS for 15 min at room temperature, followed by an additional 2 min in PBS-0.5% Triton X-100. The coverslips were incubated in a blocking solution (DUO82007) for 1 h at 37 °C. Primary antibodies were then applied and incubated overnight at 4 °C. The next day, after two washes, samples were incubated with Duolink PLA probes (anti-rabbit minus (DUO92005) and anti-mouse plus (DUO92001)) at 37 °C for 1 h, followed by the ligation and amplification using the Detection Reagents Red (DUO92008) according to the manufacturer's instructions. Nuclei were stained with DAPI, and imaging was performed using the EVOS M7000 microscope (Thermo-Fisher). The antibodies used in PLA are listed as follows: DHX9 (ab26271, abcam, 1:400), DHX9 (sc-137232, 1:600), FLAG (F7425, Merck, 1:500), S9.6 (MABE1095, Merck, 1:300), SUMO2/3 (ab3742, abcam, 1:600), DDX21 (sc-376953, Santa Cruz, 1:100), SF3B1 (sc-514655, Santa Cruz, 1:100), XRN2 (sc-365258, Santa Cruz, 1:100), Pol II (sc-47701, Santa Cruz, 1:100), and PARP1 (sc-8007, Santa Cruz, 1:100).

## DRIP for Western blot and DRIP-RT-qPCR

DRIP was performed with some modifications to the protocol as previously described[69]. HeLa cells with or without DHX9 depletion were

introduced with SFB-DHX9 variants (WT and K120R) using lipofecta-mine 2000. Two days later, cells were harvested by trypsinization, washed with ice-cold 1X PBS, and centrifuged at $300 \times g$ for 4 minutes. The resulting cell pellets were resuspended in lysis buffer (5 mM PIPES pH 7.4, 80 mM KCl, and 0.5% Igepal) at 4 °C for 30 min and centrifuged at $1000 \times g$ for 10 min to obtain the nuclei pellets. The nuclei pellets were then resuspended in the nuclear lysis buffer (25 mM Tris-HCl pH 8.0, 5 mM EDTA, and 1% SDS), incubated on ice for 30 min, followed by proteinase K (60 μg) digestion at 55 °C for 3 h. Nucleic acids were extracted using phenol/chloroform/isoamyl alcohol (25:24:1), applied to tubes containing phase-lock gel, and centrifuged at $1500 \times g$ for 10 min. The aqueous phase obtained was mixed with 1/10 volume of 3 M sodium acetate (pH 5.2) and 2 volumes of ice-cold 100% ethanol for DNA precipitation. The purified DNA was sonicated using an ultrasonicator (Covaris) to generate DNA fragments ranging from 200 to 500 bp. Fragment size was confirmed by agarose gel electrophoresis. To IP DNA/RNA hybrids, 5 μg of sheared nucleic acid was diluted in 1X DRIP binding buffer (10 mM sodium phosphate, pH 7.0, 0.14 M NaCl, and 0.05% Triton X-100) and subjected to IP using 2 μg of S9.6 antibody. The mixture was incubated overnight at 4 °C and then bound to protein G dynabeads (ThermoFisher) for 2 h. Dynabeads were subsequently washed and incubated with 140 μg of proteinase K in elution buffer (50 mM Tris, pH 8.0, 10 mM EDTA, 0.5% SDS) at 55 °C for 45 minutes to separate captured DNA/RNA hybrids. The eluates were extracted with phenol/chloroform again, and DNA was harvested using an ethanol precipitation protocol. To assess the enrichment of R-loops across the β-actin and γ-actin genes, RT-qPCR analysis was conducted with a Bio-Rad CFX Connect instrument and SYBR-Green reagents (BP170-8882, Bio-Rad). Each assay was performed in triplicate, and 10% of the sheared nucleic acid was used as input. DRIP signals at regions of β-actin and γ-actin genes were calculated as a percentage of the input and normalized to the intron 1 (in1) of β-actin and γ-actin genes, respectively. All Primers used in DRIP analysis are summarized in Supplementary Table 4.

## Slot blot assay

The slot blot assay was performed with minor modifications to a previously described protocol[23]. HeLa cells or inducible clones were collected and cell pellets were resuspended in lysis buffer (5 mM PIPES pH 7.4, 80 mM KCl, and 0.5% Igepal) at 4 °C for 30 min and centrifuged at $1000 \times g$, 4 °C for 10 min to obtain the nuclei pellets. The nuclei pellets were then resuspended in the nuclear lysis buffer (25 mM Tris-HCl pH 8.0, 5 mM EDTA, and 1% SDS), incubated on ice for 30 min, and treated with proteinase K (60 μg) at 55 °C for 3 h, followed by phenol/chloroform extraction, applied to tubes containing phase-lock gel, and centrifuged at $1500 \times g$ for 10 min. The aqueous phase obtained was mixed with 1/10 volume of 3 M sodium acetate (pH 5.2) and 2 volumes of ice-cold 100% ethanol for DNA precipitation. Purified nucleic acids (400 ng), treated with or without 5 U RNH (M0297L, NEB), were applied to a Hybond-N+ hybridization membrane (Cytiva) using a Bio-Dot SF apparatus (1706542, Bio-Rad). The membranes were cross-linked by UVC at 0.12 J/m$^2$ (UVP CL-1000) and subsequently blocked with 5% skim milk in TBS-T at room temperature for 1 h. The double-stranded DNA (dsDNA) was stained with methylene blue (50484, Sigma-Aldrich) at room temperature for 10 min. DNA/RNA hybrids were detected using the S9.6 antibody (MABE1095, Merck, 1:2000) at 4 °C overnight, followed by an HRP-conjugated secondary antibody (G21040, ThermoFisher, 1:10,000) and incubated for 1 h, followed by three additional washes in TBS-T. The signals were detected using chemiluminescence reagents and imaged by the ChemiDoc imaging system.

## Transcriptional termination readthrough assay

HeLa cells or inducible clones transfected with control or DHX9 siRNA for 24 h were subsequently transfected with SFB-DHX9 variants or

subjected to doxycycline induction, respectively. 2 d after plasmids transfection or doxycycline induction, cells were collected for RNA extraction using Direct-zol RNA MiniPrep Plus (Zymo Research) according to the manufacturer's instructions. 1 µg of total RNA was used to generate cDNA using SuperScript IV reverse transcriptase and random hexamers (ThermoFisher). The transcriptional termination assay was performed as previously described[26]. The abundance of RNA across polyA-proximal regions of $\beta$-actin and $\gamma$-actin genes was determined using qPCR. All primers used are summarized in Supplementary Table 4.

### Helicase assay

Cells were transiently transfected with SFB-DHX9 variant plasmids for 48 h and lysed in NETN buffer. The lysates were treated with Universal nuclease at room temperature for 15 min. SFB-DHX9 variant proteins were immunoprecipitated from HeLa cells using anti-FLAG magnetic beads, as described in the IP section, with washes in NETN buffers of increasing stringency (150 mM, 300 mM, 500 mM, and 750 mM NaCl). The purified SFB-DHX9[WT] was subjected to in vitro SUMOylation. Equal amounts of SUMOylated DHX9[WT] and the K120R mutant were used in helicase assays. The helicase assay, adapted from previous studies[59,70], used DNA/RNA hybrids generated by annealing 3' 6-carboxyfluorescein (FAM)-labeled DNA oligonucleotides (DNA30-FAM) with RNA oligonucleotides (RNA60) at a 1:4 molar ratio. RNA60 was synthesized using an RNA synthesis kit (E2050, NEB) with OligoRNA60-T7 DNA as a template. The RNA60 and DNA30-FAM were heated to 95 °C for 4 min and slowly cooled to room temperature. DNA/RNA hybrids (10 nM) were incubated with bead-purified SFB-DHX9[WT] and SFB-DHX9[K120R] at 37 °C for 25 min in reaction buffer (20 mM PIPES pH 7.5, 3 mM $MgCl_2$, 2 mM ATP, 2 mM DTT, and 0.1 mg/mL BSA). Proteinase K was added for an additional 8-min incubation. Half of the samples were analyzed using 10% nondenaturing polyacrylamide gels, and the FAM signal was assessed using the Typhoon biomolecular imager (cytiva). All oligonucleotide sequences are listed in Supplementary Table 5.

### Flow cytometry for Annexin V-APC analysis and γH2AX/RNH correlation assay

To monitor cell death, HeLa derivative clones were transfected with control or DHX9 siRNA for 24 h before inducing SFB-DHX9 proteins by doxycycline (200 ng/mL). 96 h after siRNA transfection, floating and attached cells were collected for the Annexin V-APC analysis (Biolegend) according to the manufacturer's instructions. To monitor the correlation of accumulated γH2AX signal and the overexpression of RNH, we used flow cytometry to determine the double-positive signal of γH2AX and RNH. HeLa derivative clones were transfected with control or DHX9 siRNA for 24 h, followed by the transfection of RNH-mCherry or the empty vector with doxycycline added simultaneously. 96 h after siRNA transfection, cells were collected and fixed, stained with anti-γH2AX, washed, and incubated with Alexa-488-conjugated secondary antibody. All data were acquired using the LSRFortessa Flow cytometer (BD Bioscience) and analyzed using Kaluza software (Beckman Coulter).

### ATP-based cell viability assay

HeLa inducible cell lines were transfected with DHX9 siRNA in 6-well plates for 24 h, followed by doxycycline induction (200 ng/mL) to induce the expression of SFB-DHX9 proteins. The next day, cells were reseeded to 96-well plates (4000 cells per well), and 20 h post-reseeding, cells were treated with DMSO, CPT, HU, cisplatin, and berzosertib for 24 h. The cell viability was determined using the Cell Titer-Glo 2.0 luminescent viability assay (Promega) following the manufacturer's instructions. The values were acquired using an ELISA reader (BioTek) and normalized to DMSO samples in each condition.

### Colony formation and cell survival assays

For colony formation assay, HeLa cells transfected with siDHX9 for 24 h were introduced with the empty vector or SFB-DHX9 variants for 24 h. Cells were then reseeded to six-well plates at a density of 500 cells per well. 10 days later, colonies were fixed with methanol and stained using 0.5% crystal violet. For cell survival assays, cells transfected with control or DHX9 siRNA for 24 h were subsequently transfected with SFB-DHX9 variants or the empty vector. 24 h post-plasmid transfection, the daily cell count was performed. For cell survival assay using HeLa inducible clones, cells transfected with control or DHX9 siRNA for 24 h were introduced with or without RNH-mCherry. 4 d after DHX9 knockdown, the cell count was performed.

### Statistics and reproducibility

Statistical tests shown in figure legends were conducted using GraphPad Prism 9 (GraphPad Software, Inc.). Each experiment was repeated at least three times unless stated otherwise. Data presentation was carried out with GraphPad Prism 9 and Affinity Designer (Serif Ltd.).

### Reporting summary

Further information on research design is available in the Nature Portfolio Reporting Summary linked to this article.

## Data availability

All data supporting the findings of this study can be found in the manuscript and its supplementary documents. The Uniprot (https://www.uniprot.org/), GPS-SUMO (https://sumo.biocuckoo.cn/), SUMO-plot (https://www.abcepta.com/sumoplot), and AlphaFold (https://alphafold.ebi.ac.uk/) databases are publicly accessible. The raw mass spectrometry data of DHX9 interactome are available via ProteomeXchange with the identifier PXD044366. Source data are provided in this paper.

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

## Acknowledgements

We thank colleagues at National Taiwan University—Drs. Jing-Jer Lin, Shu-Chun Teng, and Nei-Li Chan for insightful discussions, and Dr. Chi-Kang Tseng for providing the Typhoon biomolecular imager. We thank Dr. Kuen-Phon Wu (Academia Sinica) for providing pKWu139, pWH36, and reagents for recombinant protein purification. We thank support from the Flow Cytometric Analyzing and Sorting Core (NTU College of Medicine) and Ross Tomaino with the Taplin Mass Spectrometry Facility (Harvard Medical School) for LC-MS/MS analysis. We also thank Yuen Chen in H.-P.C. Chu's lab for technical assistance. M.-Y.L. was supported by the National Taiwan University (112L4000). This research was supported by grants from the National Science and Technology Council (110-2628-B-002-037, 111-2628-B-002-018, 112-2628-B-002-009) and National Taiwan University (CDP-111L7737, 112L7720, and 113L7704) to C.-S.P.W., and by the Center of Precision Medicine from the Featured Areas Research Center Program within the framework of Higher Education Sprout Project by the Ministry of Education (NTU-113L901402C).

## Author contributions

C.-S.P.W. conceived and supervised the study. B.-Z.Y. and M.-Y.L. performed PLA, slot blots, DRIP-qPCR, S9.6 IP, helicase assay, and co-IP. K.-L.C., B.-Z.Y., C.-A.C. and C.-S.P.W. performed IP for DHX9 SUMOylation. B.-Z.Y., M.-Y.L., K.-L.C., Y.-L.Chien., Y.-L.Chen., L.-Y.T., K.-R.L. and C.-S.P.W. performed WB. K.-R.L. contributed to MS analysis. B.-Z.Y. performed immunofluorescence staining and viability assays. M.-Y.L. contributed to in vitro SUMOylation assays. H.-P.C.C. provided reagents and essential technical assistance in DRIP-qPCR, in vitro RNA synthesis, and slot blot analysis. B.-Z.Y., M.-Y.L. and C.-S.P.W. analyzed data and wrote the manuscript with inputs from all authors.

## Competing interests

The authors declare no competing interests.
