## [Peer Review File · Nature Communications]

DHX9 SUMOylation is required for the suppression of R-loop-associated genome instabilityREVIEWER COMMENTS

Reviewer #1 (Remarks to the Author):

In this manuscript, Yang et al. investigated the sumoylation-mediated regulation of DHX9's function in suppressing R-loop-associated genome instability. They found that DHX9 is modified by SUMO2/3 at K120 by the E3 ligase PIAS4. Abolishing DHX9 sumoylation, as demonstrated by introducing K120R mutation, increases cellular R-loops and causes DNA damage, which leads to increased apoptosis and reduced cell proliferation/survival. These cellular phenotypes can be mitigated by the overexpression of RNase H, which specifically disrupts R-loops. Mechanistically, they found that sumoylation can enhance DHX9 interaction with R-loops, as well as many known R-loop interaction proteins. How these interactions translate to enhanced R-loop resolution is not known.

Overall, this is a straightforward study about how a particular PTM might affect the protein function. The results summarized in this manuscript described several observations, both at molecular and cellular levels, regarding what could happen when cells express sumoylation-deficient mutant DHX9 (K120R). However, the mechanistic insights required to better understand this PTM are missing. Specifically, 1) how K120 sumoylation enhances DHX9 R-loop resolution capacity is not clear. All the comparisons are made between WT and K120R mutant DHX9. This mutant is useful to understand the importance of K120. But without other complementary assays that directly involve sumoylated DHX9, itself revealed little information about the role of K120 sumoylation. In particular, the authors mentioned that the level of DHX9 sumoylation is very low, so the majority of WT DHX9 in cells is likely not sumoylated. This might also explain some of the dominant-negative effects of the K120R mutant in comparison to the WT DHX9. 2) Whether DHX9 sumoylation is regulated is not studied. Thus, both the mechanism of action and the biological significance of this modification remain unclear. Additionally, several key biochemical assays are missing to clearly define i) PIAS4 is the E3 ligase; ii) K120 is the site of PIAS4-mediated sumoylation; iii) sumoylation enhances DHX9 interaction with R-loops and other candidate proteins. These major critiques greatly diminished my enthusiasm.

Detailed critiques:

1. Sumo-1 modification of DHX9 by Ubc9 has been reported (Ref. 41). The site is located at N-terminal 1-137aa, covering the SUMO2/3 site K120. In Fig. 1c, the expression of SUMO1 is much lower than SUMO2. So, the authors might just miss it. Typically, defining a substrate sumoylation requires an in vitro sumoylation assay. They should perform this assay to systematically characterize SUMO1 vs. SUMO2/3 on DHX9.
2. Endogenous DHX9 sumoylation was not obvious. The data presented in Supplementary Fig. 1c is not convincing.
3. In Fig. 2a-c, although there is a clear separation of WT vs. K120R in terms of cell proliferation, the rescue expression of WT (and K120R) DHX9 in the knockdown cells only reaches about one-third of DHX9 expression in corresponding siControl cells (in U2OS cells, compare lane 5 with lane 2 in Fig. 2c). why there seems no difference in cell proliferation?
4. The apoptosis data in Fig. S2C need to be labeled clearly. The % of apoptosis is not obvious, whereas the quantification was quite dramatic (Fig. 2e). Fig. 2f, K120R-3 was not expressed in siControl cells.
5. Fig. 3a, siControl should be included to be compared to siDHX9. How to explain why K120R causes more DNA damage in comparison to the siDHX9? If this is because of dominant negative effects, why this is not reflected in cell proliferation and apoptosis assay? Also, if siDHX9 does not cause more damage than siDHX9+WT, what are the causes for the reduced proliferation? There are obvious disconnections between DNA damage and cell survival/apoptosis.
6. From Fig. 3a and supp Fig. 3a, there seems to be differences in WT vs. K120R DHX9 localization. WT DHX9 is more localized in nuclear speckles, whereas the K120R is not. There is also an inconsistent pattern of localization in response to CPT treatment. How to explain this?
7. Knockdown of DHX9 by siRNA was reported to reduce global R-loops as measured by slot-blot (Ref.

- 26) The result in Fig. 4a is in contrast to the previous publication. How to explain this?
8. Formation of R-loops at termination sites is mainly at the pause sites, and R-loops decrease dramatically once pass this site. The results presented in Fig. 4c and 4d showed strong R-loops formed downstream of the pause sites, which have not been overserved previously (ref. 26). How to explain this?
9. Fig. 5 showed that K120R reduces DHX9 interaction with R-loops and many of the RNA binding proteins. It is unclear if this is due to KR mutation or loss of sumoylation. Does knockdown PIAS4 cause the same? As mentioned before, the authors should carefully check K120R localization. Most of the listed RBPs are in nuclear speckles. K120R could just be mislocalized.
10. How does sumoylation enhance DHX9 interaction need to be investigated? Is it through direct interaction or mediated by other proteins?
11. How does sumoylation enhance DHX9 interaction need to be investigated? The authors speculate through SIM of its target proteins. This should be tested out.
12. Fig. 6a-d, what causes the initial differences in viability without Dox induction? Shouldn't they be the same, as they all siDHX9?
13. Characterization of PIAS4 as the E3 ligase should be performed with in vitro sumoylation assays. K120R should be included.
14. For the better logical flow of the manuscript, Fig. 1 & Fig. 7 should be brought together/near and Fig. 2, 3, & 6 should be brought together/near.

Reviewer #2 (Remarks to the Author):

While previous studies identified DHX9 as a regulator of R-loops, how it functions in R-loop regulation is not known. The authors provide convincing evidence that Sumoylation of DHX9 promotes R-loop homeostasis in this current manuscript under review. Specifically, they found that DHX9 is SUMOylated by PIAS4 in asynchronous cells at K120. Preventing DHX9 SUMOylation by mutating K120 to Alanine increased DNA damage and apoptosis, resulting in reduced cell proliferation. Mechanistically, DHX9-K120R mutants reduced its ability to interact with many different R-loop-interacting proteins. Overall, the results are convincing and highlight novel contribution of SUMOylation of DHX9 in R-loop regulation.

Some key questions remaining:

1. Are SUMO-DHX9 interaction with other proteins are R-loop dependent? This would get into the mechanism of how DHX9-SUMO may contribute to R-loop regulation.
2. Is DHX9 a direct substrate of PIAS3 and/or PIAS4?

Some additional experiments may strengthen the findings.

1. In Figure 3, the title is DNA double-strand breaks (DSB). However, gH2AX is technically a measure of DNA damage not DSB. Performing DNA break assays (ie. Comet assay Neutral or alkaline) would be a better measure for this.
2. Figure 1C: The author is missing a HA alone in the IP condition, which is crucial for determining the stickiness of HA plasmid to FLAG beads. Perhaps showing one of the key findings with the HA-SUMO2 alone for Flag IP would be sufficient as a control.
3. Figure 1: can the author explain the HA band in the first lane without any exogenous plasmids? The HA input in the fourth lane (SUMO2) is a lot darker than the third lane (SUMO1), can the authors explain the potential of abundance of SUMO2 and how that could potentially skew the result as a preference of SUMO2 rather than SUMO1?
4. Statistical analysis for different groups in one experiment should be done by ANOVA or others, and not Student's t-test.
5. Figure 5: The authors showed DHX9 mutant disrupted protein-protein interactions. Are these interaction R-loop dependent? Perhaps suppressing R-loops by RNaseH overexpression in cells or

RNaseH treatment in extracts in DHX9-WT IP would be informative for this.

6. Figure 7: Does PIAS3 and PIAS4 knockdown (single or double) reduce DHX9 association to R-loops?

7. Figure 7: It would be nice to know if DHX9 SUMOylation by PIAS3 or PIAS4 is direct by in vitro experiment.

Reviewer #3 (Remarks to the Author):

The work by Bing-Ze Yang et al., entitled "DHX9 SUMOylation is required for the suppression of R-loop-associated genome instability" unravels the relevance of SUMO conjugation to the RNA helicase DHX9 for R-loop metabolism and homeostasis and ultimately, for genomic stability and cell survival.

The manuscript includes a large and robust amount of experimental work, that in general has been conducted and presented in an appropriate manner. The axis of the work is convincing and the manuscript is clearly written.

I found this work to be of interest for the scientific community covering different tiers within the field of gene expression regulation, and providing novel experimental evidence to the growing body of data linking SUMO conjugation to RNA biogenesis.

Having said so, I include below a list of concerns and important experimental controls that should be addressed in order to strengthen the conclusions, to improve the quality of the work even further, and to make it suitable for publication in Nature Communications:

Specific comments:

INTRODUCTION SECTION

1)Page 2, first paragraph, where it says: "R-loops are abundant near the termination regions of polyA-dependent genes", I wonder whether may be more appropriate to refer to 3'end regions instead of termination regions, taking into account that there is no precise transcriptional termination information within eukaryotic protein-coding genes, besides the 3'end cleavage and polyadenylation signal (as far as I understand).

2)Page 3, last paragraph, I have few concerns regarding the data corresponding to PIAS4 (see further comments in point 11). By no means, the involvement of PIAS4 in R-loop homeostasis has been demonstrated in this work.

As the authors will read below, I suggest they should remove the data on PIAS4, as in their present form do not really add any relevant information. In case the authors decide to keep it (after addressing my experimental concerns), the last sentence of the introduction needs to be rephrased, unless additional data showing a clear connection between PIAS4 and DHX9-dependent R-loop homeostasis is presented. Considering the current data (and assuming the required controls are provided) what can be said is the following: PIAS4 is able to regulate SUMO conjugation to DHX9. In addition, DHX9 SUMOylation modulates R-loop homeostasis by enhancing DHX9 R-loop binding as well as its interaction with multiple R-loop modulators, thereby safeguarding genome stability.

RESULTS SECTION:

3)Figure 1:

Panel c)

Considering expression levels for HA-SUMO1 and HA-SUMO2 have never been tested (or shown) and also the fact that global SUMOylation in input lanes is so much higher with HA-SUMO2 than with HA-SUMO1, is not pertinent to compare the SUMOylation levels of DHX9 with the two SUMO variants. The only conclusion that can be taken from this analysis is that SUMO2 conjugation to DHX9 is easily detectable under the current experimental conditions.

Panel e)

It is more appropriate to say that the SUMOylation of the K120R mutant is markedly reduced or decreased, instead of "inhibited".

Panel f)

upper-right quadrant (IP Flag, WB anti SUMO2/3), is the band marked with the lowest asterisk truly specific? as far as I can see, it is also observed in the lane corresponding to transfection with empty vector.

Panel g)

It is important to show DHX9 levels upon siRNA-mediated knockdown as well as the expression levels of the different DHX9 variants upon tet-on induction, corresponding to the cells used for the PLA.

4)Figure 2,

Please, state in Materials and Methods or in the figure legend, whether the cells were seeded at the same time they were being transfected with siRNAs, as the upper time line seems to indicate.

5)Page 5,

Please note that siRNA does not lead to the truly absence of the protein but to its reduction. Revise expressions such as "In the absence of endogenous DHX9, cells expressing DHX9K120R exhibited reduced cell numbers..." or "...display increased DNA DSBs in the absence of endogenous DHX9..." Also, in this page, and related to Figure 3, I am not fully convinced by the suggestion that SFB-DHX9K120R displays a dominant negative effect. I consider that it would be appropriate to test the effect of over-expressing this mutant in the context of unperturbed levels of the endogenous protein in order to make this conclusion.

6)Figure 3

Also, it would be important to show expression levels of the transfected proteins under the tested experimental conditions.

Regarding the subcellular localization of DHX9, mentioned in the first paragraph of page 6, I consider that in order to assure that SUMOylation does not affect DHX9 localization it is required to show lower magnification images, in which several cells can be analyzed, in which cytoplasmic areas can be easily visualize and identify, and also to perform a nucleus-cytoplasm segmentation analysis.

If this is not performed, please tone down the conclusions of this section, pointing only to the observation that the K120R mutant is still able to localize in the nucleus.

Furthermore, if one or more NLS has/have been described for this protein, please, indicate the position in the scheme depicted in Figure 1a, as it would be informative to see the localization with respect to SUMOylation sites.

7)Figure 4:

Again, the controls for levels of protein over-expression and knock-down are required.

4a:

Bar-graph, please clarify whether the two first bars do correspond to control siRNA + empty vector.

4b:

Please, revise the spelling of the label "isolation of nuclei and chromatin extraction"

4c and d, legend:

Please note that it should read RT-qPCR instead of "qRT-PCR"

4e and f:

Please clarify that they correspond to RT-qPCR analysis.

Considering that the results shown in panel f are so convincing, and so are the ones shown in panel e for the "D" amplicon, what is the explanation for the lack of effect observed with the "C" amplicon in

panel e? is there any speculation to discuss? Otherwise, is it necessary to show it?

8)Figure 5:

I consider this to be the major point to revise along the whole manuscript.

In my opinion, it is somehow striking that the K120R mutant loses interaction with every protein tested. Have the authors detected any unaffected interaction of this mutant even with an R-loop unrelated protein?

In this context, it is important to mention that the use of the K76R mutant to rule out that the observed effects may be due to structural changes following K120 mutation does not seem appropriate (last paragraph of page 7). In this sense, the most pertinent control that indeed should be done in order to assure that the observed effects are related to DHX9 SUMO conjugation is to mutate the SUMO consensus sequence flanking the lysine in position 120 WITHOUT altering the particular K residue (for instance, E122A mutation). This mutant should mirror the effects observed for the K120R mutant. I would not ask the authors to repeat the all set of experiments with this new mutant, but at least few of them. That would be the more convincing result for involving SUMO conjugation and for ruling out any possible structural change and/or any other PTM that may be occurring at that particular K residue.

Of course, if the K120R mutant does NOT lose every protein-protein interaction, then, that would also be an important indicative of structural preservation.

9)Page 8, main text:

In connection with the results from Figure 5, it could be appropriate to clearly state that DHX9 SUMOylation is required for RNA-dependent and RNA-independent protein interactions.

10)Figure 6

Again, the control of expression levels of transfected and knock-down proteins is missing.

11)Figure 7;

As mentioned at the beginning of my report, I consider the results regarding PIAS4 could be remove from the work without causing any significant loss.

As pointed above (see item #8), it is striking that DHX9K120R loses interaction with PIAS3, PIAS4 and RanBP2. It loses interaction with every protein tested.

Furthermore, the rationale for analyzing the interaction of the E3 ligases with the K120R mutant is not clear to me. The fact that DHX9 SUMOylation could be regulated by any of these E3 ligases does not mean the interaction between DHX9 and these enzymes has to be mediated by the conjugated SUMO. If this were the case, how would the unmodified protein be recognized by the corresponding E3 ligase for its SUMOylation? Any comment on this would be greatly welcome.

Apart from this somehow unclear point, panel e of this figure is not convincing as there is less Flag-SFB-DHX9, both in input and in IP lane, in the presence of siRNA against PIAS4, which could be the reason for also observing less SUMOylated DHX9 (HA-SUMO-DHX9). If a statement about the regulation of SUMO conjugation levels of DHX9 by PIAS4 wants to be made, it would require a more convincing image or the quantification of several biological replicates.

What about the effect of over-expressing PIAS4, or any of the other tested E3 ligases, on SUMOylation levels of DHX9? an enhancing effect may be easier to observe.

Besides these technical issues, these results do not show a connection between PIAS4 and the effects of DHX9 SUMOylation on R-loop homeostasis and genome stability. See my comment #2

Minor comment:

1) along the whole text, the authors use the following expressions that need to be corrected:

- "the corresponding vector" should be replaced by "the corresponding empty vector"

- "...cells were transfected with the vector, SFB-DHX9wt and SFB-DHX9K120R". First of all, it should say "the corresponding empty vector" and second, it should say "OR" instead of "AND", as the transfections were always done separately.

Point-by-point response to the reviewer's comments

REVIEWER COMMENTS

Reviewer #1 (Remarks to the Author):

In this manuscript, Yang et al. investigated the sumoylation-mediated regulation of DHX9's function in suppressing R-loop-associated genome instability. They found that DHX9 is modified by SUMO2/3 at K120 by the E3 ligase PIAS4. Abolishing DHX9 sumoylation, as demonstrated by introducing K120R mutation, increases cellular R-loops and causes DNA damage, which leads to increased apoptosis and reduced cell proliferation/survival. These cellular phenotypes can be mitigated by the overexpression of RNase H, which specifically disrupts R-loops. Mechanistically, they found that sumoylation can enhance DHX9 interaction with R-loops, as well as many known R-loop interaction proteins. How these interactions translate to enhanced R-loop resolution is not known.

Overall, this is a straightforward study about how a particular PTM might affect the protein function. The results summarized in this manuscript described several observations, both at molecular and cellular levels, regarding what could happen when cells express sumoylation-deficient mutant DHX9 (K120R). However, the mechanistic insights required to better understand this PTM are missing. Specifically, 1) how K120 sumoylation enhances DHX9 R-loop resolution capacity is not clear. All the comparisons are made between WT and K120R mutant DHX9. This mutant is useful to understand the importance of K120. But without other complementary assays that directly involve sumoylated DHX9, itself revealed little information about the role of K120 sumoylation. In particular, the authors mentioned that the level of DHX9 sumoylation is very low, so the majority of WT DHX9 in cells is likely not sumoylated. This might also explain some of the dominant-negative effects of the K120R mutant in comparison to the WT DHX9. 2) Whether DHX9 sumoylation is regulated is not studied. Thus, both the mechanism of action and the biological significance of this modification remain unclear. Additionally, several key biochemical assays are missing to clearly define i) PIAS4 is the E3 ligase; ii) K120 is the site of PIAS4-mediated sumoylation; iii) sumoylation enhances DHX9 interaction with R-loops and other candidate proteins. These major critiques greatly diminished my enthusiasm.

We thank the reviewer for her/his constructive comments to improve our manuscript. In our revised manuscript, we have incorporated a new set of experiments, including *in vitro* SUMOylation assays, additional mutations in the LKAE consensus SUMOylation motif, and identification of SIMs in DHX9-interacting proteins, as suggested by the reviewer. Furthermore, our rescue experiments using SUMO2-fused DHX9^{K120R} clearly demonstrate the functional importance of DHX9 SUMOylation. We hope our new results and clarifications will convince the reviewer that DHX9 SUMOylation at K120 plays an important role in R-loop balance and genome stability.

Detailed critiques:

1. Sumo-1 modification of DHX9 by Ubc9 has been reported (Ref. 41). The site is located at N-terminal 1-137aa, covering the SUMO2/3 site K120. In Fig. 1c, the expression of SUMO1 is much lower than SUMO2. So, the authors might just miss it. Typically, defining a substrate sumoylation requires an *in vitro* sumoylation assay. They should perform this assay to systematically characterize SUMO1 vs. SUMO2/3 on DHX9.

We thank the reviewer's comments. In our experiments testing SUMO conjugation by co-expressing HA-SUMO and SFB-DHX9, we utilized a sufficient amount of HA-SUMO (3 µg DNA per 10 cm culture dish) to minimize non-specific SUMO conjugation signals. It is important to note that the previous study by Argasinska et al., 2004 (**now Ref. 42, formerly Ref. 41**) used 4 times more HA-SUMO1 DNA than our experiment.

Following the reviewer's suggestions, we have repeated the experiments and conducted *in vitro* SUMOylation assays. This time, we used freshly made HA-SUMO plasmids with a slightly higher amount of HA-SUMO1 (4 µg DNA per 10cm culture dish) plasmid compared to HA-SUMO2 (3 µg DNA per 10cm culture dish), as detailed in the new legend. This approach was adopted to counteract the potential for lower global HA-SUMO1 signals observed in the **original Fig. 1c**. As shown in the **new Fig. 1c**, the global HA-SUMO1 signal in input samples was slightly stronger than HA-SUMO2. However, the HA-SUMO1 signal associated with SFB-DHX9 immunoprecipitated under denaturing conditions was much weaker than the HA-SUMO2 signal.

Second, we conducted *in vitro* SUMOylation assays using a commercial kit (abcam) to further validate DHX9's SUMOylation (the **new Fig. 1d, h**). We noted that the DHX9 fragment (1-399) is predominantly conjugated by SUMO2, in contrast to the much weaker SUMO1 conjugation. While our data do not rule out the possibility of DHX9 being modified by SUMO1, as indicated by the weaker SUMO1 signal, they suggest that DHX9 is preferentially conjugated by SUMO2 over SUMO1 at K120.

2. Endogenous DHX9 sumoylation was not obvious. The data presented in Supplementary Fig. 1c is not convincing.

We have overcome technical challenges in examining endogenous DHX9 SUMOylation and have updated **Supplementary Fig. 1c**.

3. In Fig. 2a-c, although there is a clear separation of WT vs. K120R in terms of cell proliferation, the rescue expression of WT (and K120R) DHX9 in the knockdown cells only reaches about one-third of DHX9 expression in corresponding siControl cells (in U2OS cells, compare lane 5 with lane 2 in Fig. 2c). why there seems no difference in cell proliferation?

Although the ectopically expressed SFB-DHX9^{WT} in lane 5 was less than in lane 2 of **Fig. 2c**, the total amount of DHX9 (SFB-DHX9 + residual endogenous DHX9) is close to the level of endogenous DHX9 in U2OS cells (lane 1, siControl with vector). This amount may be sufficient for cellular proliferation. To ensure that our conclusions are not dependent on a single cell line or analysis, we also performed additional assays with HeLa cells or HeLa stable cell lines, as shown in other panels of **Fig. 2**.

4. The apoptosis data in Fig. S2C need to be labeled clearly. The % of apoptosis is not obvious, whereas the quantification was quite dramatic (Fig. 2e). Fig. 2f, K120R-3 was not expressed in siControl cells.

We thank the reviewer for her/his suggestions and questions. The representative Annexin V-APC staining data are now clearly labeled in the **new Supplementary Fig. 2C**, with a revised legend that specifies what was quantified. Regarding the **original Fig. 2f**, SFB-DHX9^{K120R} in the K120R-3 clone transfected with siControl was indeed expressed, albeit at a very low level. We have updated this result with a longer exposure in the **new Fig. 2f**.

5. Fig. 3a, siControl should be included to be compared to siDHX9. How to explain why K120R causes

more DNA damage in comparison to the siDHX9? If this is because of dominant negative effects, why this is not reflected in cell proliferation and apoptosis assay? Also, if siDHX9 does not cause more damage than siDHX9+WT, what are the causes for the reduced proliferation? There are obvious disconnections between DNA damage and cell survival/apoptosis.

We would like to point out that the experimental endpoint for **the original Fig. 3a** differed from that of the survival/apoptosis assays shown in **Figs. 2d and 2e**. In the **original Fig. 3a**, cells were fixed for immunostaining three days post-knockdown, while the survival/apoptosis assays were conducted at 96 hours post-knockdown. Thus, it is not appropriate to compare the DNA damage in the **original Fig. 3a** to the survival/cell death in **Fig. 2d and 2e**.

On the other hand, we would like to apologize for the confusion caused by the different experimental timelines across different assays. To ensure consistency and prevent misleading interpretations, we have conducted experiments for the **new Fig. 3** using the same timeline as in Fig. 2d. Similarly, the **new Supplementary Fig. 3** was conducted following the timeline used in Figs. 2a and 2b. All cells were fixed for immunostaining at the endpoints, and control siRNA groups were included as recommended by the reviewer. Additionally, the expression levels of the induced or transfected SFB-DHX9 variants have been detailed in the **new Fig. 3f** and **Supplementary Fig. 3f**.

As shown in the **new Fig. 3**, we observed no disconnection between DNA damage and cell survival/death. Additionally, in the **new Fig. 3** and **Supplementary Fig. 3**, there were no discrepancies in DNA damage levels between siDHX9+Vect and siDHX9+K120R. Regarding the increased DNA damage caused by K120R compared to siDHX9+Vect, as observed in the **original Fig. 3** (cells fixed on day 3 after knockdown), we speculate that K120R may impair genome stability through mechanisms that are not yet understood. It is also important to note that HeLa cells depleted of DHX9 for 4 days were largely lost during immunofluorescence staining following CPT treatment, preventing the examination of the impact of DHX9^{K120R} on CPT-induced γ H2AX foci formation under this condition.

6. From Fig. 3a and supp Fig. 3a, there seems to be differences in WT vs. K120R DHX9 localization. WT DHX9 is more localized in nuclear speckles, whereas the K120R is not. There is also an inconsistent pattern of localization in response to CPT treatment. How to explain this?

We thank the reviewer for the question. We would like to clarify that the representative images in the original Fig. 3a were a single plane of Z-stacked images obtained from an LSM880 confocal microscope (Zeiss), while the images in the original Supplementary Fig. 3a were taken with a fluorescence microscope (EVOSM7000, Thermo). In response to the reviewer's question, we thoroughly reviewed our images. From both the confocal and lower magnification images, we did not observe significant differences in DHX9 patterns between WT and K120R cells (**Fig. R1**). Upon CPT treatment, both WT and K120R cells exhibited increased RING-like patterns around or inside the nucleoli (**Fig. R1**), a phenomenon also noted in CPT-treated cells in Ref. 45. Overall, we observed no marked differences in these patterns. We apologize for any confusion caused by the selection of representative images.

Fig. R1. Representative images of SFB-DHX9 staining (anti-FLAG) in HeLa inducible cell lines treated with DMSO or CPT (original Fig. 3a). The left panel is images taken by a confocal microscope (60X). The right panel images are taken by EVOSM7000 (40X).

7. Knockdown of DHX9 by siRNA was reported to reduce global R-loops as measured by slot-blot (Ref. 26) The result in Fig. 4a is in contrast to the previous publication. How to explain this?

We thank the reviewer for pointing this out. The slot blot analysis shown in **Fig. 4a** represents data from three independent experiments, conducted by two researchers. We detected the R-loop signals using the S9.6 antibody and also stained for total dsDNA from duplicate samples as a loading control. In contrast, **Ref. 26** only presents the S9.6 slot blot signals without a loading control (Fig. 3E of **Ref. 26**). While the siRNA used in both our study and Ref. 26 is the same, other factors such as cellular context or sample preparation may account for the observed differences. Furthermore, several recent studies carried out with different cell types have shown that knockdown of DHX9 leads to an increase in R-loops, either globally or at specific genomic regions (Yuan W, et al. *Nuclei Acids Res* 2021, Huang N, et al. *iScience* 2023, Patel PS, *J Clin Invest.* 2021). Perhaps employing a combination of DRIP-seq and MapR for a more thorough examination globally could yield additional insights to explain these discrepancies.

8. Formation of R-loops at termination sites is mainly at the pause sites, and R-loops decrease dramatically once pass this site. The results presented in Fig. 4c and 4d showed strong R-loops formed downstream of the pause sites, which have not been overserved previously (ref. 26). How to explain this?

We were aware of the noted difference when conducting the DRIP-qPCR analysis of R-loops in **Fig. 4c and 4d**. Our data in these figures were obtained using the standard protocol described in Ref. 65 and were repeated at least three times independently by two researchers. The HeLa cell line, purchased directly from the ATCC, was maintained under standard procedures and routinely checked using the MycoAlert Mycoplasma Detection Kit (Lonza) to ensure no contamination. We would like to point out that in the same study, a strong signal of R-loops was detected further downstream of the pause site (siDHX9 in Fig. 5D,

Ref. 26), a phenomenon that differs from what was shown in Fig. 4H of Ref. 26. Given that factors such as cellular context, technical sensitivity, and experimental variations may influence the differences in R-loop signals observed across studies, employing a combination of DRIP-seq and MapR for a more thorough examination could yield additional insights to explain these discrepancies.

9. Fig. 5 showed that K120R reduces DHX9 interaction with R-loops and many of the RNA binding proteins. It is unclear if this is due to KR mutation or loss of sumoylation. Does knockdown PIAS4 cause the same? As mentioned before, the authors should carefully check K120R localization. Most of the listed RBPs are in nuclear speckles. K120R could just be mislocalized.

We thank the reviewer for the insightful questions and suggestions. To validate whether the deficiency seen in K120R was due to KR mutation or loss of SUMOylation, we mutated additional residues in the LKAE (Ψ KxE/D) SUMOylation motif of DHX9. Our **new data in Fig. 6e, f** confirmed that the E122A mimics the effect of K120R, disrupting its association with multiple RNA processing proteins and R-loops, while A121G does not. SUMOylation analysis of these DHX9 variants further confirmed that E122A abolishes K120-SUMOylation, as shown in the **new Fig. 6g**. Furthermore, as shown in **new Fig. 7f**, silencing UBC9, a SUMO2 conjugase, resulted in reduced interactions between DHX9 and several RNA-interacting proteins. Our new results support the importance of DHX9 SUMOylation in the interactions with its target proteins.

Regarding the results of PIAS4, please refer to point 13, where the full response is addressed.

Response to the localization of K120R is addressed in point 6.

10. How does sumoylation enhance DHX9 interaction need to be investigated? Is it through direct interaction or mediated by other proteins?

To examine the direct interaction between SUMOylated DHX9 and R-loop-binding proteins like PARP1, DDX21, or other proteins, we would need SUMOylated DHX9 and recombinant or purified proteins of interest for *in vitro* co-IP. This is technically challenging and requires considerable time and effort. Additionally, such *in vitro* interactions might not be detectable if other post-translational modifications also play a role. Given that PARP1 is known to associate with DHX9 from several studies, and proteins involved in RNA processing, including DDX21, have been identified in global SUMO-binding studies (**new Refs. 48 and 49**), we decided to first investigate whether DHX9 interacts with R-loop-binding proteins via SUMO-SIM interactions. We identified SIMs in PARP1 and DDX21 and showed that DHX9 enhances its interaction with PARP1 and DDX21 via SUMO-SIM as described in the results (**new Fig. 7 and 8**).

11. How does sumoylation enhance DHX9 interaction need to be investigated? The authors speculate through SIM of its target proteins. This should be tested out.

We thank the reviewer for this insightful comment. PARP1 and DDX21 have been uncovered from studies of global non-covalent SUMO interaction networks (**new Ref.48, 49**), suggesting the presence of SIMs in these proteins. Following the reviewer's questions, we investigated if DHX9 interacts with PARP1 and DDX21 via SUMO-SIM interactions. Mutations in these SIMs markedly reduced their interactions with DHX9, as shown in **new Fig. 7a-e and Supplementary Fig. 7d**. Furthermore, as shown in **new Fig. 7f**, silencing UBC9, a SUMO2 conjugase, resulted in reduced interactions between DHX9 and several RNA-interacting proteins. Our new results suggest that the SUMO-SIM interaction is crucial for interactions of DHX9 with its target proteins.

12. Fig. 6a-d, what causes the initial differences in viability without Dox induction? Shouldn't they be the same, as they all siDHX9?

We thank the reviewer for the questions. The panels **a-d** in the **original Fig. 6** were from a single trial, which was inadvertently misplaced during manuscript preparation. We have corrected these with data from three independent experiments in the **new Fig. 5a-d** and sincerely apologize for this mistake. In general, only occasional cases exhibited a significant difference in cell viability in the DOX-minus groups, as shown in the **new Fig. 5c and Table R1**. Given that the viability experiments were conducted in triplicate and independently repeated three times, the differences observed in a particular treatment in DOX-minus groups were within reasonable experimental variation. The statistical results of samples without DOX induction are shown in Table R1.

Cisplatin Conc	Summary	P Value
0 Row 1:WT_DOX- vs. Row 1:K120R_DOX-	ns	>0.9999
0.25 Row 2:WT_DOX- vs. Row 2:K120R_DOX-	*	0.0212
0.5 Row 3:WT_DOX- vs. Row 3:K120R_DOX-	ns	0.4815
1 Row 4:WT_DOX- vs. Row 4:K120R_DOX-	ns	>0.9999
2 Row 5:WT_DOX- vs. Row 5:K120R_DOX-	ns	>0.9999
4 Row 6:WT_DOX- vs. Row 6:K120R_DOX-	ns	>0.9999
HU Conc	Summary	P Value
0 Row 1:WT_DOX- vs. Row 1:K120R_DOX-	ns	>0.9999
0.125 Row 2:WT_DOX- vs. Row 2:K120R_DOX-	ns	0.3288
0.25 Row 3:WT_DOX- vs. Row 3:K120R_DOX-	****	<0.0001
0.5 Row 4:WT_DOX- vs. Row 4:K120R_DOX-	ns	>0.9999
1 Row 5:WT_DOX- vs. Row 5:K120R_DOX-	ns	>0.9999
Perzoesertib Conc	Summary	P Value
0 Row 1:WT_DOX- vs. Row 1:K120R_DOX-	ns	>0.9999
0.125 Row 2:WT_DOX- vs. Row 2:K120R_DOX-	ns	0.9819
0.25 Row 3:WT_DOX- vs. Row 3:K120R_DOX-	ns	0.4027
0.5 Row 4:WT_DOX- vs. Row 4:K120R_DOX-	ns	0.349
1 Row 5:WT_DOX- vs. Row 5:K120R_DOX-	ns	0.0651
CPT Conc	Summary	P Value
0 Row 1:WT_DOX- vs. Row 1:K120R_DOX-	ns	>0.9999
12.5 Row 2:WT_DOX- vs. Row 2:K120R_DOX-	**	0.0032
25 Row 3:WT_DOX- vs. Row 3:K120R_DOX-	ns	>0.9999
50 Row 4:WT_DOX- vs. Row 4:K120R_DOX-	ns	0.8993

Table R1. Statistical Results for the Comparison of WT and K120R in the Doxycycline-Minus Group.

13. Characterization of PIAS4 as the E3 ligase should be performed with *in vitro* sumoylation assays. K120R should be included.

We thank the reviewer for the question. We here addressed the full response regarding PIAS4 results raised by this reviewer in points 5 and 13. In the **original Fig. 7c**, knocking down PIAS3 did not affect DHX9's SUMOylation, while PIAS4 knockdown led to a moderate reduction in DHX9 SUMOylation. Therefore, we conducted *in vitro* SUMOylation with a commercial SUMOylation kit and bacterially purified PIAS4. Our new data revealed that adding recombinant PIAS4 enhances DHX9 SUMOylation by SUMO2 (**Fig. R2**), suggesting that PIAS4 can enhance DHX9 SUMOylation by SUMO2 *in vitro*.

However, it is important to highlight new observations that prompted us to reconsider the role of PIAS4 in this manuscript. First, the knockdown of either PIAS4 alone or both PIAS3 and PIAS4 did not result in the same protein interaction defects observed with DHX9 K120R (**Fig. R3**). Second, neither PIAS4 nor PIAS3/4 knockdown affected the association between DHX9 and R-loops (**Fig. R4**), phenomena we cannot currently explain. Identifying the true E3 ligase responsible for the SUMOylation of DHX9 will take a considerable of time and effort. We think it would be more suitable for future study. Therefore, we have

removed the results concerning E3 ligases and focused our study on understanding how DHX9-SUMO enhances protein-protein interactions. Accordingly, we have added **new Figs. 7 and 8**, demonstrating that DHX9 SUMOylation is crucial for its interactions with PARP1 and DDX21 via the SUMO-SIM interaction.

14. For the better logical flow of the manuscript, Fig. 1 & Fig. 7 should be brought together/near and Fig. 2, 3, & 6 should be brought together/near.

We thank the reviewer for this comment. We have rearranged the figures as suggested. The **original Fig. 6** is now the **new Fig. 5**. For the **original Fig. 7**, we have removed it and updated it with new data, as addressed in **point 13**.

Reviewer #2 (Remarks to the Author):

While previous studies identified DHX9 as a regulator of R-loops, how it functions in R-loop regulation is not known. The authors provide convincing evidence that Sumoylation of DHX9 promotes R-loop homeostasis in this current manuscript under review. Specifically, they found that DHX9 is SUMOylated by PIAS4 in asynchronous cells at K120. Preventing DHX9 SUMOylation by mutating K120 to Alanine increased DNA damage and apoptosis, resulting in reduced cell proliferation. Mechanistically, DHX9-K120R mutants reduced its ability to interact with many different R-loop-interacting proteins. Overall, the results are convincing and highlight novel contribution of SUMOylation of DHX9 in R-loop regulation.

We sincerely thank this reviewer for her/his overall positive comments and constructive comments to our manuscript. Below, we have carefully addressed each of the questions raised by this reviewer.

Some key questions remaining:

1. Are SUMO-DHX9 interaction with other proteins are R-loop dependent? This would get into the mechanism of how DHX9-SUMO may contribute to R-loop regulation.

We thank the reviewer for this insightful comment. Following the reviewer's question, we added **new Fig. 6i**, demonstrating that overexpressing RNH-mCherry in cells does not disrupt the interactions between DHX9 and R-loop-interacting proteins, such as PARP1 and DDX21. Additionally, as shown in the **original Fig. 5k and Supplementary Fig. 5g** (now **new Supplementary Fig. 6k, I**), RNH treatments in PLA analysis did not eliminate the interactions of DHX9 with PARP1 or DDX21. In the **revised Fig. 7**, we further confirmed the existence of SUMO-interacting motifs (SIMs) in PARP1 and DDX21, and provided new evidence showing that these SIMs are crucial for their interactions with DHX9 via SUMO-SIM interactions. Our new results suggest that interactions of DHX9 with R-loop-binding proteins are likely independent of R-loops.

2. Is DHX9 a direct substrate of PIAS3 and/or PIAS4?

We thank the reviewer for this critical question. The full response to this question is addressed in point 7.

Some additional experiments may strengthen the findings.

1. In Figure 3, the title is DNA double-strand breaks (DSB). However, gH2AX is technically a measure of DNA damage not DSB. Performing DNA break assays (ie. Comet assay Neutral or alkaline) would be a better measure for this.

We thank the reviewer for pointing this out. We have toned down by correcting the text, stating "DNA damage" instead of "DNA DSB."

2. Figure 1C: The author is missing a HA alone in the IP condition, which is crucial for determining the stickiness of HA plasmid to FLAG beads. Perhaps showing one of the key findings with the HA-SUMO2 alone for Flag IP would be sufficient as a control.

We thank the reviewer for raising concerns about the potential stickiness of the HA plasmid to FLAG beads. Although HA plasmid alone was not included in our DHX9 SUMOylation analysis experiments, several pieces of evidence rule out the possibility that the observed SUMO-DHX9 signals were due to non-specific interactions between HA and FLAG beads: 1) Immunoprecipitation of SFB-DHX9 from cells transfected with SFB-DHX9 variants confirmed the conjugation of endogenous SUMO2/3 in WT, but not in K120R and 2KR variants. 2) In the **new Fig. 1c**, DHX9-SUMO signals were more prominent with HA-

SUMO2 than with HA-SUMO1. 3) As shown in the **new Fig. 1f (formerly Fig. 1e)**, HA-SUMO2 combined with SFB-DHX9 K120R or 2KR, but not with WT, exhibited a marked reduction in DHX9-SUMO signals. 4) The newly added *in vitro* SUMOylation assays further validate the SUMOylation of DHX9 (**new Fig. 1d, h**).

3. Figure 1: can the author explain the HA band in the first lane without any exogenous plasmids? The HA input in the fourth lane (SUMO2) is a lot darker than the third lane (SUMO1), can the authors explain the potential of abundance of SUMO2 and how that could potentially skew the result as a preference of SUMO2 rather than SUMO1?

Since the reviewer did not specify which panel in the **original Fig. 1**, we speculated that it was panel C. The HA band in the first lane without any plasmids, seen in the lower INPUT panel, could be due to non-specific detection, which we sometimes observed when a fresh or higher concentration of HA antibody was used. Also, such a non-specific band was not detected in the **original Fig. 1d, e (now new Fig. 1e, f)**. Since an equal amount of HA-SUMO1 and HA-SUMO2 DNA was used in the original Fig. 1c, the more abundant SUMO2 signal could be due to uneven SUMO expression or global SUMO conjugation.

In our experiments testing SUMO conjugation by co-expressing HA-SUMO and SFB-DHX9, we utilized a sufficient amount of HA-SUMO (3 µg DNA per 10 cm culture dish) to minimize non-specific SUMO conjugation signals. While the other group used 4 times more HA-SUMO1 DNA than our experiment (Ref. 42). To confirm DHX9 SUMOylation, we repeated the experiment with freshly made HA-SUMO plasmids. This time, we used a slightly higher amount of HA-SUMO1 plasmid compared to HA-SUMO2, as detailed in the new legend. This approach was adopted to counteract the potential for lower global HA-SUMO1 signals observed in the **original Fig. 1c**.

As shown in the **new Fig. 1c**, the global HA-SUMO1 signal in input samples was slightly stronger than HA-SUMO2. However, the HA-SUMO1 signal on SFB-DHX9 immunoprecipitated under denaturing conditions was much weaker than the HA-SUMO2 signal.

Second, we performed *in vitro* SUMOylation assays using a commercial SUMOylation kit (abcam) to further validate the SUMOylation of DHX9 (**new Fig. 1d, h**). We observed that the DHX9 fragment (1-399) is markedly conjugated by SUMO2, while SUMO1 conjugation is much weaker. Our data do not exclude the possibility of DHX9 being modified by SUMO1, as we can observe a weaker signal of SUMO1. Rather, our results suggest that DHX9 is preferentially conjugated by SUMO2 over SUMO1 at K120.

4. Statistical analysis for different groups in one experiment should be done by ANOVA or others, and not Student's t-test.

We thank the reviewer for pointing out this issue. We have redone and corrected the statistical analysis as suggested and updated the statistical analysis section in Methods and figure legends.

5. Figure 5: The authors showed DHX9 mutant disrupted protein-protein interactions. Are these interaction R-loop dependent? Perhaps suppressing R-loops by RNaseH overexpression in cells or RNaseH treatment in extracts in DHX9-WT IP would be informative for this.

Following the reviewer's suggestion, we added **new Fig. 6i**, which demonstrates that overexpressing RNH-mCherry in cells does not markedly disrupt the interactions between DHX9 and R-loop-interacting proteins, such as PARP1 and DDX21. Additionally, as shown in the **original Fig. 5k and Supplementary Fig. 5g (now new Supplementary Fig. 6k, l)**, RNH treatments in PLA analysis did not eliminate the interactions of DHX9 with PARP1 or DDX21.

In the **revised Fig. 7**, we further confirmed the existence of SUMO-interacting motifs (SIMs) in PARP1 and DDX21, and provided new evidence showing that these SIMs are crucial for their interactions with DHX9 via SUMO-SIM interactions. Our new results suggest that interactions of DHX9 with R-loop-binding proteins are likely independent of R-loops.

6. Figure 7: Does PIAS3 and PIAS4 knockdown (single or double) reduce DHX9 association to R-loops?

Following the reviewer's critical question, we knocked down PIAS3, PIAS4, or PIAS3/4 double knockdown and examined the association between DHX9 and R-loops using PLA analysis. The results showed that neither PIAS4 nor PIAS3/4 knockdown affected the association between DHX9 and R-loops (Fig. R4). Please refer to point 7 for a more detailed response regarding PIAS3/4 results.

Fig. R4. The effects of knocking down SUMO E3 ligases on the association between DHX9 and R-loops. HeLa cells transfected with siPIAS3, siPIAS4, or both siPIAS3/4 for 3 d were subjected to PLA analysis using the indicated antibodies. Protein levels were determined by Western blot with the indicated antibodies.

7. Figure 7: It would be nice to know if DHX9 SUMOylation by PIAS3 or PIAS4 is direct by *in vitro* experiment.

In the **original Fig. 7c**, knocking down PIAS3 did not affect DHX9's SUMOylation, while PIAS4 knockdown led to a moderate reduction in DHX9 SUMOylation. Therefore, we conducted *in vitro* SUMOylation with a commercial SUMOylation kit and bacterially purified PIAS4. Our new data revealed that adding recombinant PIAS4 enhances DHX9 SUMOylation by SUMO2 (**Fig. R2**), suggesting that PIAS4 can enhance DHX9 SUMOylation by SUMO2 *in vitro*.

However, it is important to highlight new observations that prompted us to reconsider the role of PIAS4 in this context. First, the knockdown of either PIAS4 alone or both PIAS3 and PIAS4 did not result in the same protein interaction defects observed with DHX9 K120R (**Fig. R3**). Second, neither PIAS4 nor PIAS3/4 knockdown affected the association between DHX9 and R-loops (**Fig. R4**), phenomena we cannot currently explain. Identifying the true E3 ligase responsible for the SUMOylation of DHX9 will take considerable time and effort. We think it would be more suitable for future study. Therefore, we have removed the results concerning E3 ligases and refocused our study on understanding how DHX9-SUMO enhances protein-protein interactions. Accordingly, we have added **new Figs. 7 and 8**, demonstrating that DHX9 SUMOylation is crucial for its interactions with PARP1 and DDX21 via the SUMO-SIM interaction.

Reviewer #3 (Remarks to the Author):

The work by Bing-Ze Yang et al., entitled “DHX9 SUMOylation is required for the suppression of R-loop-associated genome instability” unravels the relevance of SUMO conjugation to the RNA helicase DHX9 for R-loop metabolism and homeostasis and ultimately, for genomic stability and cell survival.

The manuscript includes a large and robust amount of experimental work, that in general has been conducted and presented in an appropriate manner. The axis of the work is convincing and the manuscript is clearly written.

I found this work to be of interest for the scientific community covering different tiers within the field of gene expression regulation, and providing novel experimental evidence to the growing body of data linking SUMO conjugation to RNA biogenesis.

Having said so, I include below a list of concerns and important experimental controls that should be addressed in order to strengthen the conclusions, to improve the quality of the work even further, and to make it suitable for publication in Nature Communications:

We thank the reviewer for his/her support and positive suggestions for improving our manuscript. Below, we have carefully addressed each of the questions raised by this reviewer.

Specific comments:

INTRODUCTION SECTION

1)Page 2, first paragraph, where it says: “R-loops are abundant near the termination regions of polyA-dependent genes”, I wonder whether may be more appropriate to refer to 3'end regions instead of termination regions, taking into account that there is no precise transcriptional termination information within eukaryotic protein-coding genes, besides the 3'end cleavage and polyadenylation signal (as far as I understand).

We thank the reviewer for pointing this out. The original texts were based on findings from a study mapping R-loop distribution across genomic regions. In agreement with the reviewer's suggestion and for greater accuracy, we have now rephrased the sentence as follows: “R-loops are abundant near the 3'-end regions of polyA-dependent genes”, which is highlighted in red in the main text.

2)Page 3, last paragraph, I have few concerns regarding the data corresponding to PIAS4 (see further comments in point 11). By no means, the involvement of PIAS4 in R-loop homeostasis has been demonstrated in this work.

As the authors will read below, I suggest they should remove the data on PIAS4, as in their present form do not really add any relevant information. In case the authors decide to keep it (after addressing my experimental concerns), the last sentence of the introduction needs to be rephrased, unless additional data showing a clear connection between PIAS4 and DHX9-dependent R-loop homeostasis is presented. Considering the current data (and assuming the required controls are provided) what can be said is the following: PIAS4 is able to regulate SUMO conjugation to DHX9. In addition, DHX9 SUMOylation modulates R-loop homeostasis by enhancing DHX9 R-loop binding as well as its interaction with multiple R-loop modulators, thereby safeguarding genome stability.

We sincerely thank the reviewer's comments regarding the data on PIAS4 in the original submission. As addressed in point 11, we removed the PIAS4 results as suggested.

RESULTS SECTION:

3)Figure 1:

Panel c)

Considering expression levels for HA-SUMO1 and HA-SUMO2 have never been tested (or shown) and also the fact that global SUMOylation in input lanes is so much higher with HA-SUMO2 than with HA-SUMO1, is not pertinent to compare the SUMOylation levels of DHX9 with the two SUMO variants. The only conclusion that can be taken from this analysis is that SUMO2 conjugation to DHX9 is easily detectable under the current experimental conditions.

We thank the reviewer for pointing this out. To confirm DHX9 SUMOylation, we carefully repeated this experiment with freshly made HA-SUMO plasmids. Experimentally, we used a slightly higher amount of HA-SUMO1 plasmid compared to HA-SUMO2, as detailed in the new legend. This approach was adopted to counteract the potential for lower global HA-SUMO1 signals observed in the **original Fig. 1c**.

As shown in the **new Fig. 1c**, the global HA-SUMO1 signal in input samples was slightly stronger than HA-SUMO2. However, the HA-SUMO1 signal on SFB-DHX9 immunoprecipitated under denaturing conditions was much weaker than the HA-SUMO2 signal.

Second, we performed *in vitro* SUMOylation assays using a commercial SUMOylation kit from abcam to further validate the SUMOylation of DHX9 (**new Fig. 1d, h**). We observed that the DHX9 fragment (1-399) is markedly conjugated by SUMO2, while SUMO1 conjugation is much weaker.

Our *in vivo* and *in vitro* data do not exclude the possibility of DHX9 being modified by SUMO1. Rather, they suggest that DHX9 is preferentially conjugated by SUMO2 over SUMO1 at K120.

Panel e)

It is more appropriate to say that the SUMOylation of the K120R mutant is markedly reduced or decreased, instead of "inhibited".

We have corrected this in the main text.

Panel f)

upper-right quadrant (IP Flag, WB anti SUMO2/3), is the band marked with the lowest asterisk truly specific? as far as I can see, it is also observed in the lane corresponding to transfection with empty vector.

Former Fig. 1f (now new Fig. 1g)

The reasons we believe the band marked with the lowest asterisk represents a true SUMO signal are:

- I. The band with the lowest asterisk in WT and K76R samples is much stronger than those in K120R and 2KR samples.
- II. Throughout our study, we consistently observed residual SUMO2/3 signals in K120R and 2KR samples (**Fig. 1f, g, and new Fig. 6g**), which were never completely eliminated.

Therefore, the band appearing in the empty vector sample is likely a non-specific detection in this particular blot.

Panel g)

It is important to show DHX9 levels upon siRNA-mediated knockdown as well as the expression levels of the different DHX9 variants upon tet-on induction, corresponding to the cells used for the PLA.

We thank the reviewer for highlighting this issue. For PLA analysis, we consistently prepared duplicate samples to verify protein levels, but these were not included in the original submission. We have now added **new Fig. 1k**, corresponding to **new Fig. 1i, j (former Fig. 1g)**.

4)Figure 2,

Please, state in Materials and Methods or in the figure legend, whether the cells were seeded at the same time they were being transfected with siRNAs, as the upper time line seems to indicate.

We thank the reviewer for the suggestion. RNAi was reverse transfected into cancer cells. The detailed information has been updated in Methods and the figure legend (**Fig. 2a, b**).

5)Page 5,

Please note that siRNA does not lead to the truly absence of the protein but to its reduction. Revise expressions such as “In the absence of endogenous DHX9, cells expressing DHX9K120R exhibited reduced cell numbers...” or “...display increased DNA DSBs in the absence of endogenous DHX9...” Also, in this page, and related to Figure 3, I am not fully convinced by the suggestion that SFB-DHX9K120R displays a dominant negative effect. I consider that it would be appropriate to test the effect of over-expressing this mutant in the context of unperturbed levels of the endogenous protein in order to make this conclusion.

We thank the reviewer for pointing out the wording issues. We have corrected those issues in the main text. For detailed information regarding the former and **new Fig. 3**, please refer to point 6.

6)Figure 3

Also, it would be important to show expression levels of the transfected proteins under the tested experimental conditions.

Regarding the subcellular localization of DHX9, mentioned in the first paragraph of page 6, I consider that in order to assure that SUMOylation does not affect DHX9 localization it is required to show lower magnification images, in which several cells can be analyzed, in which cytoplasmic areas can be easily visualize and identify, and also to perform a nucleus-cytoplasm segmentation analysis.

If this is not performed, please tone down the conclusions of this section, pointing only to the observation that the K120R mutant is still able to localize in the nucleus.

Furthermore, if one or more NLS has/have been described for this protein, please, indicate the position in the scheme depicted in Figure 1a, as it would be informative to see the localization with respect to SUMOylation sites.

We apologize for any misunderstanding caused by the wording and fully agree with this comment. We did not perform a nucleus-cytoplasm segmentation analysis and have toned down the conclusion by revising the text to state, “*The observations were not attributed to the failure of nuclear localization of DHX9^{K120R}, nor to different expression levels of DHX9^{WT} or DHX9^{K120R} in HeLa inducible lines (related to **new Fig. 3a, f**).*”

To ensure consistency and prevent misleading interpretations, we have conducted experiments for the **new Fig. 3** using the same timeline as in Fig. 2d. Similarly, the **new Supplementary Fig. 3** was conducted following the timeline used in Figs. 2a and 2b. All cells were fixed for immunostaining at the endpoints, and control siRNA groups were included as recommended by the reviewer. Additionally, the expression levels of the induced or transfected SFB-DHX9 variants have been detailed in the **new Fig. 3f** and **Supplementary Fig. 3f**.

Regarding the NLS, we have updated the **new Fig. 1a** to depict the location of the NLS at the C-terminal region of DHX9.

7)Figure 4:

Again, the controls for levels of protein over-expression and knock-down are required.

We have added new data in **Supplementary Fig. 4b, c** to show the expression levels of transfected SFB-DHX9 variants and the endogenous DHX9.

4a:

Bar-graph, please clarify whether the two first bars do correspond to control siRNA + empty vector.

4b:

Please, revise the spelling of the label “isolation of nuclei and chromatin extraction”

4c and d, legend:

Please note that it should read RT-qPCR instead of “qRT-PCR”

4e and f:

Please clarify that they correspond to RT-qPCR analysis.

We thank the reviewer for pointing these out. We have **updated Fig. 4a, b** and corrected the text in the legend of **Fig. 4c, d**. The transcriptional termination readthrough analysis related to **Fig. 4e, f** was originally described in the Methods section. As suggested by the reviewer, we have also updated the main text and the legends to clarify that they correspond to the RT-qPCR analysis.

Considering that the results shown in panel f are so convincing, and so are the ones shown in panel e for the “D” amplicon, what is the explanation for the lack of effect observed with the “C” amplicon in panel e? is there any speculation to discuss? Otherwise, is it necessary to show it?

We examined the effects of DHX9^{K120R} on transcriptional readthrough downstream of the polyA site, building on a previous study (Ref. 26). To ensure comprehensive data presentation, we included all results in their entirety.

8)Figure 5:

I consider this to be the major point to revise along the whole manuscript.

In my opinion, it is somehow striking that the K120R mutant loses interaction with every protein tested. Have the authors detected any unaffected interaction of this mutant even with an R-loop unrelated protein?

In this context, it is important to mention that the use of the K76R mutant to rule out that the observed effects may be due to structural changes following K120 mutation does not seem appropriate (last paragraph of page 7). In this sense, the most pertinent control that indeed should be done in order to assure that the observed effects are related to DHX9 SUMO conjugation is to mutate the SUMO consensus sequence flanking the lysine in position 120 WITHOUT altering the particular K residue (for instance, E122A mutation). This mutant should mirror the effects observed for the K120R mutant. I would not ask the authors to repeat the all set of experiments with this new mutant, but at least few of them. That would be the more convincing result for involving SUMO conjugation and for ruling out any possible structural change and/or any other PTM that may be occurring at that particular K residue.

Of course, if the K120R mutant does NOT lose every protein-protein interaction, then, that would also be

an important indicative of structural preservation.

We sincerely thank the reviewer for the insightful comments. Regarding the **original Fig. 5, now updated to Fig. 6**, our decision to include K76R was based on two major reasons: 1) K76 was identified as a SUMO2 modification site in a proteomic study (Ref. 43), and 2) K76 is located in the same disordered linker loop as K120. Considering that the K76R mutation did not affect DHX9 SUMOylation, it serves as a useful control for demonstrating the defects caused by the K120R mutation. As summarized in the **new Supplementary Fig. 6c (formerly Fig. 5c)**, major proteins coprecipitated with endogenous DHX9 were nucleic acid-binding proteins. We tested almost all available antibodies against those proteins and failed to detect any interaction proteins unaffected by K120R.

We also fully agree with the reviewer's comments regarding the mutation of additional residues in the LKAE (Ψ KxE/D) SUMOylation motif of DHX9. Our **new data in Fig. 6e, f** confirmed that the E122A mimics the effect of K120R, disrupting its association with multiple RNA processing proteins and R-loops, while A121G does not. SUMOylation analysis of these DHX9 variants further confirmed that E122A abolishes K120-SUMOylation, as shown in the **new Fig. 6g**.

9)Page 8, main text:

In connection with the results from Figure 5, it could be appropriate to clearly state that DHX9 SUMOylation is required for RNA-dependent and RNA-independent protein interactions.

We thank the reviewer's suggestion. The sentence connected to the results from **new Fig. 6 (former Fig. 5)** has been revised as suggested.

10)Figure 6

Again, the control of expression levels of transfected and knock-down proteins is missing.

The **former Fig. 6** has been updated to the **new Fig. 5**. Data on protein expression levels are now included in the new Supplementary Fig. 5, which shows the protein levels of the samples tested in the **original Fig. 6a-d (new Fig. 5a-d)**. These cell lysates were originally collected from a duplicate set during the sensitivity assay.

11)Figure 7;

As mentioned at the beginning of my report, I consider the results regarding PIAS4 could be removed from the work without causing any significant loss.

We thank and fully agree with the reviewer's comments. Regarding the PIAS4 results, we have now removed it as suggested. The detailed reasons are addressed in the last paragraph.

As pointed above (see item #8), it is striking that DHX9K120R loses interaction with PIAS3, PIAS4 and RanBP2. It loses interaction with every protein tested.

Furthermore, the rationale for analyzing the interaction of the E3 ligases with the K120R mutant is not clear to me. The fact that DHX9 SUMOylation could be regulated by any of these E3 ligases does not mean the interaction between DHX9 and these enzymes has to be mediated by the conjugated SUMO. If this were the case, how would the unmodified protein be recognized by the corresponding E3 ligase for its SUMOylation? Any comment on this would be greatly welcome.

We were also surprised that DHX9^{K120R} failed to interact with all tested proteins. The detailed response to this has been addressed in item 8. Initially, we did not know if any of these E3 ligases coprecipitated with

DHX9 were involved in the SUMOylation of DHX9 or any proteins associated with DHX9. Therefore, we included K120R in the PLA assays. The **original Fig. 7** has been replaced with our new findings in the **new Fig. 7 and 8**, showing DHX9 SUMOylation is crucial for its interactions with PARP1 and DDX21 via the SUMO-SIM interaction.

Apart from this somehow unclear point, panel e of this figure is not convincing as there is less Flag-SFB-DHX9, both in input and in IP lane, in the presence of siRNA against PIAS4, which could be the reason for also observing less SUMOylated DHX9 (HA-SUMO-DHX9). If a statement about the regulation of SUMO conjugation levels of DHX9 by PIAS4 wants to be made, it would require a more convincing image or the quantification of several biological replicates.

What about the effect of over-expressing PIAS4, or any of the other tested E3 ligases, on SUMOylation levels of DHX9? an enhancing effect may be easier to observe.

Besides these technical issues, these results do not show a connection between PIAS4 and the effects of DHX9 SUMOylation on R-loop homeostasis and genome stability. See my comment #2

We thank the reviewer for raising concerns about our PIAS4 results. We admit that, at the present time, our limited data do not fully support the role of PIAS4 in this story. Although our additional data showed that bacterially expressed PIAS4 enhances *in vitro* SUMOylation of DHX9 by SUMO2, knocking down PIAS4 did not exhibit the same defects observed in DHX9^{K120R} (Fig. R2, 3). Whether there is a redundancy of E3 ligases involved or PIAS4 is not the E3 that directly regulates DHX9 remains unclear. Figuring out the true E3 ligase responsible for the SUMOylation of DHX9 will take a considerable of time and effort, which we think would be more suitable for future study.

Minor comment:

1) along the whole text, the authors use the following expressions that need to be corrected:

-“the corresponding vector” should be replaced by “the corresponding empty vector”

-“...cells were transfected with the vector, SFB-DHX9wt and SFB-DHX9K120R”. First of all, it should say “the corresponding empty vector” and second, it should say “OR” instead of “AND”, as the transfections

were always done separately.

We thank the reviewer for pointing out these issues. We have corrected those in the whole text.

REVIEWER COMMENTS

Reviewer #1 (Remarks to the Author):

Summary

In the revised manuscript, the authors addressed many of my concerns, and the manuscript is clearly improved. However, there remains one key question unanswered from my previous review comments – how does SUMO2 promote DHX9 function in R-loop resolution? They showed that loss of sumoylation reduces DHX9 interaction with R-loop and R-loop-associated proteins. All the results provided to support this conclusion are indirect – CoIP, PLA et al. They need to test if sumoylation affects DHX9's activity in binding to (affinity) and resolving (activity) R-loops using EMSA (or fluorescence polarization assay) and in vitro helicase assay with R-loop substates. Also, it is unclear how reduced interaction with R-loop-associated proteins would impact R-loop resolution by DHX9. Are these proteins involved in recruiting DHX9 to R-loop sites? Is there less DHX9 on chromatin when its interaction with these proteins is inhibited? If yes, does the recruitment require sumoylation? Characterization of how sumoylation promotes DHX9-mediated R-loop resolution is the key conclusion of this manuscript and currently, it is unclear.

The authors use SUMO2 fused DHX9 K120R as a way to restore sumoylated DHX9 in cells. As the authors mentioned, only a small portion of DHX9 is SUMO2 modified, thus expressing the fusing construct should provide fully sumoylated DHX9. One would expect a full rescue or even enhanced protein-protein interaction (if this fusing construct faithfully recapitulates DHX9 sumoylation). However, this is not the case for both PARP1 and DDX21 interactions (Fig. 8a and Supp Fig. 8a), both of which only change marginally. How to explain this? Also, will it be able to restore the normal level of R-loops?

The authors removed their initial results on PIAS4 due to its less relevance in DHX9 sumoylation in vivo. It is my opinion that identifying the E3 ligase and characterizing the regulation of DHX9 sumoylation is essential for this manuscript to be published in Nature Communications.

Reviewer #2 (Remarks to the Author):

The revised version of the manuscript has addressed all my points/concerns. The new PARP1 and DDX21 SIM motifs are quite interesting and strengthen the findings.

One minor point that would be nice to have included is the recent paper published by the authors showing DHX9 phosphorylation also plays a role in R-loop regulation (Liu et al, Nucleic Acids Research 2024). It would be nice to see how the authors speculate the phosphorylation and this new SUMOylation in the discussion. No additional data required.

Reviewer #3 (Remarks to the Author):

I am happy to see that the authors made a big effort to answer most reviewers concern, adding relevant and new experimental data. Many of my previous comments have been addressed and I consider the quality of the paper has been greatly improved.

However, I still have some additional comments to make, most of them related to the way the concepts and the conclusions are being enunciated and just one related to a piece of recently added experimental data (the SUMO fusion to DXH9):

1) ABSTRACT, line 20: to my knowledge, there is no certainty about the DHX9 K120R mutant

completely lacking SUMO conjugation (even not seeing a signal in a WB assay does not guaranty absolute absence of a given protein moiety). Moreover, the authors comment in their point-by-point response that they always observed "residual" SUMO2/3 signal in K120R and 2KR mutants", suggesting that it may very well exist other SUMOylation sites within this protein. After going through the whole manuscript, I found a sentence indicating so within the Discussion section (line 378): "the K120R mutation in DHX9 doesn't eliminate all SUMO conjugation signals, hinting at additional modification sites".

Therefore, I suggest the authors to be more careful with their statements (along the whole text) and just say that the mutant has diminished or reduced SUMO conjugation (or SUMOylation), instead of stating: "The DHX9 K120R mutant, lacking SUMO2/3 conjugation.....".

2) line 121: please indicate between brackets that DMSO is the vehicle.

3) line 125: please include a comment or refer to other section/figure of the paper indicating that DNA damage was indeed experimentally verified, upon the selected treatments (CPT and HU).

4) lines 139-141: the in vitro results with the wild type protein fragment (DHX9 1-399) does not give any information about the SUMOylation at a particular residue (K120). Only upon using the K120R mutant fragment, the authors can conclude that it is preferably conjugated by SUMO2 rather than SUMO1 at K120. Therefore, revised the way this paragraph is written. May be, it should say "The results showed again that DHX9 1-399 was preferably conjugated by SUMO2 rather than SUMO1 at K120, as introducing the K120R mutation to this DHX9 fragment markedly reduced its SUMOylation in vitro".

5) line 149: Considering the authors have only analyzed K79 and K120 (or E122) mutants in their experiments, I do not consider appropriate to conclude that K120R is "the primary" SUMO2/3 conjugation site within this protein. I suggest to be more cautious and just conclude that these findings strongly support K120 as a bonafide (or as a major, or as an important) SUMO conjugation site of DHX9.

6) line 289-290: "To ensure the observed effects were directly linked to DHX9SUMO, we mutated additional residues in the LKAE SUMOylation motif". The way this sentence is enunciated leads to think that additional mutations have been added to the K120 mutant, which is not the case. Clarify that the SUMO acceptor lysine remains unchanged in this mutant, and only a residue within the SUMOylation consensus motif has been replaced, disrupting this motif and therefore expectedly perturbing SUMO conjugation at K120.

7) along the text, the authors use the nomenclature "DHX9-SUMO" when refereeing to behavior, activities or effects caused by over-expression of the wild type protein in comparison to the SUMOylation-deficient mutant K120R (For instance: line 312 "These findings suggest that DHX9SUMO plays a critical role in binding R-loops"; lines 316-317: "The discovery that DHX9SUMO, unlike DHX9K120R, binds to various R-loop interacting proteins suggests that these interactions are SUMOylation dependent", and many others).

As far as I understand, when transfecting the wild type version of DHX9, they are dealing not only with SUMO-conjugated forms of this protein but also with the unconjugated one, as the authors have not performed any experiment with the conjugated form in isolation. Therefore, I consider more pertinent to refer always to the wild type protein, as the one that is subjected to SUMO conjugation and de-conjugation cycles, in comparison to the SUMOylation-deficient version. Furthermore, it is broadly known in the SUMO field that the proportion of modified vs unmodified molecules for a pool of a given protein at steady state is quite small. However, and despite its transient nature, this modification can still have profound consequences in the "life" of the protein and these consequences can even remain after the modification has been removed (de-conjugated). (the so called "SUMO enigma" clearly and elegantly explained by Ron Hay in "SUMO: A History of Modification", Molecular Cell 2005)

8) Figure 6g, please revise the labeling on the right side of the panel corresponding to IP (Flag). To be consistent with previous figures, I guess the labeling should be "HA (SFB-DHX9-HA-S2)".

9) Figure 7

The results presented for the interaction between DHX9 and SIM-mutated DDX21 (d and e panels) are far less convincing than the ones corresponding to DHX9 and SIM-mutated PARP1 (In this last pair of interactors, the drastic reduction observed in panel c mitigates the slight reduction observed in panel b). Please, toned down the conclusions corresponding to PARP1, including the wording for the subheading of this section. Alternatively, the authors may want to replace panel e by an image of a co-IP experiment showing a more robust diminished interaction, or may include the quantification of several assays as supplementary information.

10) Figure 8 and last subheading of the Results section:

Honestly, and despite the fact the strategy has been previously used, I must say that I am not fully convinced by the idea of fusing a linear SUMO to one end of the protein of interest in order to validate the involvement of SUMO conjugation. In particular when the analyzed SUMO acceptor site is hundreds of aminoacids away from the end of the protein.

We all know that SUMO is an approx 11 kDa peptide, that most of the time (to our current knowledge) is conjugated to internal residues forming bulky (short or long) branches in the structure of the modified protein. A priori, we cannot consider that the 3D structure obtained by the lineal fusion of one SUMO at one end of the protein sequence would be equivalent to the native 3D structure of the properly SUMO-conjugated protein (in particular at residue 120). We have learned that protein structure (form) and function are strictly and deeply connected. Having said so, I must admit that I got surprised by the success of the authors, that did indeed observe that SUMO fusion at the N-terminus of the protein is able to rescue some of the effects observed by the lack of SUMO conjugation at K120. Does this mean that SUMO attached to any part of the protein works as a bridge for tethering/recruiting other factors? If so, SUMOylation at any other lysine residue (different from K120) could also be sufficient? If so, how that is explained in the context of the observation pointed out by the authors that the K120 mutant does not lose all its SUMOylation, but still is defective in most of its activities/interactions?

Have the authors looked at the predicted structure of SUMO-fused DHX9 compared to the K120 SUMOylated DHX9? How these two structures compare in terms of the localization/exposure of the SUMO moiety within the context of the whole protein? How the addition of SUMO, at the N-term vs at K120 modifies the 3D structure of the protein?

The authors have mentioned the use of AlphaFold for analyzing SIMs. Could this or other in silico approach be used for the above asked comparison?

If the requested structural comparison has not or cannot be done, at least I consider this is a conceptual topic that fully requires to be included along the Discussion section, also referring to other studies in which the strategy of SUMO fusion has been used or discussed.

In my modest opinion, this last piece of data does not add any significant information to the already strong built story. If I were the one to take the final decision, I would remove it from the manuscript. However, I leave it up to the Editor's decision, which will be based also in the opinion of the other reviewers, whether to keep it or remove it for the final version of the paper.

11) Discussion:

line 370: please revise expressions such as "Lack of DHX9 SUMOylation"... As already mentioned, we can only assure that the K120R or E122A mutants have diminished SUMOylation or eventually, that they lack SUMOylation at K residue 120, but is not equivalent to say that the DHX9 mutants used in this study entirely lack SUMO conjugation.

Even the authors mentioned few lines below along the discussion section (line 378) that "the K120R mutation in DHX9 doesn't eliminate all SUMO conjugation signals, hinting at additional modification sites". Therefore, keep this concept clear along the whole manuscript.

Point-by-point response to the reviewer's comments

REVIEWER COMMENTS

Reviewer #1 (Remarks to the Author):

Summary

In the revised manuscript, the authors addressed many of my concerns, and the manuscript is clearly improved. However, there remains one key question unanswered from my previous review comments – how does SUMO2 promote DHX9 function in R-loop resolution? They showed that loss of sumoylation reduces DHX9 interaction with R-loop and R-loop-associated proteins. All the results provided to support this conclusion are indirect – CoIP, PLA et al. They need to test if sumoylation affects DHX9's activity in binding to (affinity) and resolving (activity) R-loops using EMSA (or fluorescence polarization assay) and in vitro helicase assay with R-loop substates. Also, it is unclear how reduced interaction with R-loop-associated proteins would impact R-loop resolution by DHX9. Are these proteins involved in recruiting DHX9 to R-loop sites? Is there less DHX9 on chromatin when its interaction with these proteins is inhibited? If yes, does the recruitment require sumoylation? Characterization of how sumoylation promotes DHX9-mediated R-loop resolution is the key conclusion of this manuscript and currently, it is unclear.

We thank the reviewer for her/his constructive comments. Over the past few months, our collaborator and we have attempted to purify the full-length recombinant DHX9 for in vitro assays. Unfortunately, we could not obtain soluble full-length DHX9. Similar challenges have been documented, regardless of whether bacterial or insect cell recombinant protein expression systems were employed (Lee et al., *Acta Crystallogr D Struct Biol* 2023). This obstacle prevents us from carrying out further biochemical analysis.

We also want to emphasize that studies on human DHX9 and *Drosophila* MLE have demonstrated that the N-terminal regions (1–150aa) of DHX9/MLE are not required for helicase activity, and the L1 linker loop, which connects dsRBD1 and 2, does not contribute to RNA binding (Izzo et al., *Nucleic Acids Res* 2008; Prabu et al., *Mol Cell* 2015; Lee et al., *Acta Crystallogr D Struct Biol* 2023). Furthermore, the K120 SUMOylation site, located within the L1 linker of DHX9, is not present in MLE. These findings suggest that the SUMOylation of human DHX9 is unlikely to be essential for its helicase activity or RNA binding capability.

Recent studies have shown that RNA-dependent DEAD-box ATPases (DDXs) regulate RNA-containing phase-separated organelles across prokaryotes and eukaryotes, forming compartments that manage RNA-processing steps (Hondele et al., *Nature* 2019). R-loop proximal proteomics has identified numerous DDXs and DHXs proteins as key to R-loop binding, highlighting the importance of various RNA processing factors in preventing dysregulated R-loop accumulation (Mosler et al., *Nat Commun* 2021). It is speculated that these R-loop-resolving proteins act together for an effective response to dysregulated R-loops, facilitated by SUMO-SIM interactions that boost protein-protein interactions for efficient R-loop resolution.

DHX9 is recognized for its association with RNA Polymerase II, playing a role in the co-transcriptional regulation of mRNA. Unlike some RNA helicases directed to R-loops by PARP1, DHX9's chromatin binding is independent of PARP1 (Cristini et al., *Cell Rep* 2018). Given the numerous proteins that interact with DHX9 on the chromatin, identifying a single protein responsible for recruiting DHX9 to the chromatin can be extremely difficult. Our findings suggest that SUMOylation is pivotal for the interactions between DHX9 and other RNA processing factors, allowing DHX9 to stably associate with R-loops or chromatin. We have this statement included in the discussion section.

The authors use SUMO2 fused DHX9 K120R as a way to restore sumoylated DHX9 in cells. As the authors mentioned, only a small portion of DHX9 is SUMO2 modified, thus expressing the fusing construct should provide fully sumoylated DHX9. One would expect a full rescue or even enhanced protein-protein interaction (if this fusing construct faithfully recapitulates DHX9 sumoylation). However, this is not the case for both PARP1 and DDX21 interactions (Fig. 8a and Supp Fig. 8a), both of which only change marginally. How to explain this? Also, will it be able to restore the normal level of R-loops?

We want to point out that slower-migrating bands observed in DHX9-SUMOylation blots (**Figs. 1c, g, and Suppl Fig. 1c**) suggest the formation of poly-SUMO2/3 chains on DHX9 K120. Due to technical issues during the last manuscript revision, we only obtained and examined the effects of 1xS2-DHX9^{K120R}. It is possible that this single SUMO2 fusion might not fully recapitulate the protein interactions seen with native SUMOylation.

Even if the improvement of DHX9's interaction with individual proteins appears marginal, the cumulative effects within the R-loop processing complex could be sufficient enough to counteract the DHX9^{K120R} defects. While we have not directly verified if S2-DHX9^{K120R} restores R-loops to normal levels, the observed increase in R-loop binding and cell viability with S2-DHX9^{K120R} implies that SUMO2 fusion can potentially restore R-loop levels close to normal.

The authors removed their initial results on PIAS4 due to its less relevance in DHX9 sumoylation in vivo. It is my opinion that identifying the E3 ligase and characterizing the regulation of DHX9 sumoylation is essential for this manuscript to be published in Nature Communications.

The SUMO E3 ligases tend to be less selective compared to that of ubiquitin E3. The depletion of PIAS4, for instance, might activate alternative pathways compensating for the defects caused by decreased DHX9 SUMOylation. Furthermore, SUMOylation can be executed by E1 and E2 enzymes alone, without requiring an E3 ligase. Whether PIAS4 or a specific SUMO E3 ligase is indeed responsible for SUMOylating DHX9 in cells remains to be determined. Identification and validation of a SUMO E3 responsible for DHX9 SUMOylation will require a substantial amount of time and resources. We hope that the reviewer would understand the potential obstacles and agree that these studies are more appropriate for a follow-up study.

References

- Cristini A, Groh M, Kristiansen MS, Gromak N. RNA/DNA Hybrid Interactome Identifies DXH9 as a Molecular Player in Transcriptional Termination and R-Loop-Associated DNA Damage. *Cell Rep.* 2018 May 8;23(6):1891-1905.
- Izzo A, Regnard C, Morales V, Kremmer E, Becker PB. Structure-function analysis of the RNA helicase maleless. *Nucleic Acids Res.* 2008 Feb;36(3):950-62
- Hondele M, Sachdev R, Heinrich S, Wang J, Vallotton P, Fontoura BMA, Weis K. DEAD-box ATPases are global regulators of phase-separated organelles. *Nature.* 2019 Sep;573(7772):144-148.
- Lee YT, Sickmier EA, Grigoriu S, Castro J, Boriack-Sjodin PA. Crystal structures of the DExH-box RNA helicase DHX9. *Acta Crystallogr D Struct Biol.* 2023 Nov 1;79(Pt 11):980-991.
- Mosler T, Conte F, Longo GMC, Mikicic I, Kreim N, Möckel MM, Petrosino G, Flach J, Barau J, Luke B, Roukos V, Beli P. R-loop proximity proteomics identifies a role of DDX41 in transcription-associated genomic instability. *Nat Commun.* 2021 Dec 16;12(1):7314.

Prabu JR, Müller M, Thomae AW, Schüssler S, Bonneau F, Becker PB, Conti E. Structure of the RNA Helicase MLE Reveals the Molecular Mechanisms for Uridine Specificity and RNA-ATP Coupling. *Mol Cell*. 2015 Nov 5;60(3):487-99.

Reviewer #2 (Remarks to the Author):

The revised version of the manuscript has addressed all my points/concerns. The new PARP1 and DDX21 SIM motifs are quite interesting and strengthen the findings.

One minor point that would be nice to have included is the recent paper published by the authors showing DHX9 phosphorylation also plays a role in R-loop regulation (Liu et al, Nucleic Acids Research 2024). It would be nice to see how the authors speculate the phosphorylation and this new SUMOylation in the discussion. No additional data required.

We thank the reviewer for her/his overall support to our revised work and the insightful suggestion. Our recent work shows that ATR-phosphorylated DHX9 at S321 plays a crucial role in preventing R-loop accumulation following genotoxic stress. Contrarily, SUMOylation at K120 occurs under normal conditions and isn't induced by HU or CPT, suggesting that SUMOylation is happening before the phosphorylation. The interplay between DHX9's SUMOylation and phosphorylation in R-loop management presents an interesting area for research. Due to limited personnel and resources, this aspect remained unexplored. Our understanding suggests SUMOylated DHX9, associated with RNA Pol II and other RNA processing factors, plays a critical role in managing R-loop levels during transcription. Upon genotoxic stress, DHX9's phosphorylation facilitates its association at transcription-replication conflicts but still requires its SUMOylation to link with R-loop resolving proteins. Whether and how the SUMOylation and phosphorylation of DHX9 are interplayed will be further investigated in a future study. Following the reviewer's comment, we have updated the discussion section accordingly.

Reviewer #3 (Remarks to the Author):

I am happy to see that the authors made a big effort to answer most reviewers concern, adding relevant and new experimental data. Many of my previous comments have been addressed and I consider the quality of the paper has been greatly improved.

However, I still have some additional comments to make, most of them related to the way the concepts and the conclusions are being enunciated and just one related to a piece of recently added experimental data (the SUMO fusion to DXH9):

We thank the reviewer for the overall support to our revised manuscript and the constructive comments. Below, we have carefully addressed each of the reviewer's comments.

1) ABSTRACT, line 20: to my knowledge, there is no certainty about the DHX9 K120R mutant completely lacking SUMO conjugation (even not seeing a signal in a WB assay does not guaranty absolute absence of a given protein moiety). Moreover, the authors comment in their point-by-point response that they always observed "residual" SUMO2/3 signal in K120R and 2KR mutants", suggesting that it may very well exist other SUMOylation sites within this protein. After going through the whole manuscript, I found a sentence indicating so within the Discussion section (line 378): "the K120R mutation in DHX9 doesn't eliminate all SUMO conjugation signals, hinting at additional modification sites".

Therefore, I suggest the authors to be more careful with their statements (along the whole text) and just say that the mutant has diminished or reduced SUMO conjugation (or SUMOylation), instead of stating: "The DHX9 K120R mutant, lacking SUMO2/3 conjugation.....".

Following the reviewer's suggestion, we have carefully revised the statement concerning the SUMOylation status of DHX9 K120R to be more precise. K120R eliminates SUMO conjugation specifically at K120. Hence, we have clarified this by stating, 'The DHX9K120R mutant lacks SUMOylation at K120.

2) line 121: please indicate between brackets that DMSO is the vehicle.

We have now corrected this to keep the consistency of the writing.

3) line 125: please include a comment or refer to other section/figure of the paper indicating that DNA damage was indeed experimentally verified, upon the selected treatments (CPT and HU).

We thank the reviewer for pointing this out. A new Western blot image showing the phosphorylation of RPA32 at serine 33 following treatment with HU or CPT has been added to **Fig. 1e** to verify the induction of the DNA damage response.

4) lines 139-141: the in vitro results with the wild type protein fragment (DHX9 1-399) does not give any information about the SUMOylation at a particular residue (K120). Only upon using the K120R mutant fragment, the authors can conclude that it is preferably conjugated by SUMO2 rather than SUMO1 at K120. Therefore, revised the way this paragraph is written. May be, it should say "The results showed again that DHX9 1-399 was preferably conjugated by SUMO2 rather than SUMO1 at K120, as introducing the K120R mutation to this DHX9 fragment markedly reduced its SUMOylation in vitro".

In Fig. 1h of the previously revised manuscript, the K120R DHX9 fragment was indeed included for comparison of SUMO conjugation. While SUMO1 conjugation was barely detectable in both WT and K120R, the SUMO2 conjugation signal was significantly observed in WT but was drastically reduced in K120R. We have now changed the text as suggested by the reviewer.

5) line 149: Considering the authors have only analyzed K79 and K120 (or E122) mutants in their experiments, I do not consider appropriate to conclude that K120R is “the primary” SUMO2/3 conjugation site within this protein. I suggest to be more cautious and just conclude that these findings strongly support K120 as a bonafide (or as a major, or as an important) SUMO conjugation site of DHX9.

We have corrected this by stating K120 as a major SUMOylation site of DHX9. Similar corrections have been made to the subheadings of Fig. 1 and suppl Fig. 1.

6) line 289-290: “To ensure the observed effects were directly linked to DHX9SUMO, we mutated additional residues in the LKAE SUMOylation motif”. The way this sentence is enunciated leads to think that additional mutations have been added to the K120 mutant, which is not the case. Clarify that the SUMO acceptor lysine remains unchanged in this mutant, and only a residue within the SUMOylation consensus motif has been replaced, disrupting this motif and therefore expectedly perturbing SUMO conjugation at K120.

Thank you. We have rephrased the sentence to clarify that K120 remains unchanged in A121G and E122A mutants.

7) along the text, the authors use the nomenclature “DHX9-SUMO” when refereeing to behavior, activities or effects caused by over-expression of the wild type protein in comparison to the SUMOylation-deficient mutant K120R (For instance: line 312 “These findings suggest that DHX9SUMO plays a critical role in binding R-loops”; lines 316-317: “The discovery that DHX9SUMO, unlike DHX9K120R, binds to various R-loop interacting proteins suggests that these interactions are SUMOylation dependent”, and many others). As far as I understand, when transfecting the wild type version of DHX9, they are dealing not only with SUMO-conjugated forms of this protein but also with the unconjugated one, as the authors have not performed any experiment with the conjugated form in isolation. Therefore, I consider more pertinent to refer always to the wild type protein, as the one that is subjected to SUMO conjugation and de-conjugation cycles, in comparison to the SUMOylation-deficient version. Furthermore, it is broadly known in the SUMO field that the proportion of modified vs unmodified molecules for a pool of a given protein at steady state is quite small. However, and despite its transient nature, this modification can still have profound consequences in the “life” of the protein and these consequences can even remain after the modification has been removed (de-conjugated). (the so called “SUMO enigma” clearly and elegantly explained by Ron Hay in “SUMO: A History of Modification”, Molecular Cell 2005)

We thank the reviewer's comment. Initially, “DHX9^{SUMO}” was used to denote either “DHX9 SUMOylation” or to indicate the wild-type DHX9, depending on the context. To enhance clarity, we have replaced “DHX9^{SUMO}” with “DHX9 SUMOylation” to specify the SUMOylation status, and “DHX9^{WT}” to specifically denote the wild-type protein in the revised manuscript.

8) Figure 6g, please revise the labeling on the right side of the panel corresponding to IP (Flag). To be consistent with previous figures, I guess the labeling should be “HA (SFB-DHX9-HA-S2)”.

Following the reviewer's suggestion, we have revised the labeling in Fig. 6g to keep the consistency with previous figures.

9) Figure 7

The results presented for the interaction between DHX9 and SIM-mutated DDX21 (d and e panels) are far

less convincing that the ones corresponding to DHX9 and SIM-mutated PARP1 (In this last pair of interactors, the drastic reduction observed in panel c mitigates the slight reduction observed in panel b). Please, toned down the conclusions corresponding to PARP1, including the wording for the subheading of this section. Alternatively, the authors may want to replace panel e by an image of a co-IP experiment showing a more robust diminished interaction, or may include the quantification of several assays as supplementary information.

Following the reviewer's comments, we have replaced panel e with new images and included the quantification results beside the Western blots in the newly revised Fig. 7. The updated data demonstrate that PARP1 and DDX21 sim mutants interact with DHX9 less efficiently than their WT counterparts. We have also carefully revised both the subheading and the content of this section to ensure our conclusions are not overstated.

10) Figure 8 and last subheading of the Results section:

Honestly, and despite the fact the strategy has been previously used, I must say that I am not fully convinced by the idea of fusing a linear SUMO to one end of the protein of interest in order to validate the involvement of SUMO conjugation. In particular when the analyzed SUMO acceptor site is hundreds of aminoacids away from the end of the protein.

We all know that SUMO is an aprox 11 KDa peptide, that most of the time (to our current knowledge) is conjugated to internal residues forming bulky (short or long) branches in the structure of the modified protein. A priori, we cannot consider that the 3D structure obtained by the lineal fusion of one SUMO at one end of the protein sequence would be equivalent to the native 3D structure of the properly SUMO-conjugated protein (in particular at residue 120). We have learned that protein structure (form) and function are strictly and deeply connected. Having said so, I must admit that I got surprised by the success of the authors, that did indeed observe that SUMO fusion at the N-terminus of the protein is able to rescue some of the effects observed by the lack of SUMO conjugation at K120. Does this mean that SUMO attached to any part of the protein works as a bridge for tethering/recruiting other factors? If so, SUMOylation at any other lysine residue (different from K120) could also be sufficient? If so, how that is explained in the context of the observation pointed out by the authors that the K120 mutant does not loses all its SUMOylation, but still is defective in most of its activities/interactions?

It is now widely recognized that a major function of SUMO conjugation is to provide a binding platform for other proteins via the SUMO-SIM interaction. Previous studies have investigated the interaction of SIM with both linear SUMO fusions and native Lys11-linked SUMO, finding almost no difference in binding (Keusekotten et al., *Biochem J* 2014; Xu et al., *Nat Commun* 2014). Additionally, linear head-to-tail SUMO fusions have been acknowledged as a suitable model for native poly-SUMOs and have been utilized in the identification of SUMO-binding proteins (Aguilar-Martinez et al., *Proc Natl Acad Sci USA* 2015; Barroso-Gomila et al., *Nat Commun* 2021).

Furthermore, fusing a SUMO moiety or linear head-to-tail poly-SUMO to a target protein, where native SUMOylation is disrupted, has become a widely used approach to investigate protein SUMOylation functions (Maison et al., *Nat Genet* 2011; Wen et al., *Cell Rep* 2017; Wu et al., *Genes Dev* 2014; Zhang et al., *Nat Commun* 2022; Zhu et al., *Mol Cell* 2023). Whether these SUMO fusions can functionally mimic native SUMOylation largely depends on the 3D protein structure of the SUMO-fused proteins.

In our study, the DHX9 K120R mutation exhibited several defects, despite the observed residual SUMOylation signal. Nonetheless, our findings suggest that the absence of SUMO on K120 is responsible for the impaired interactions with R-loop binding proteins, indicating that the residual SUMO signals are insufficient to compensate for the loss of SUMOylation at K120.

Have the authors looked at the predicted structure of SUMO-fused DHX9 compared to the K120 SUMOylated DHX9? How these two structures compare in terms of the localization/exposure of the SUMO moiety within the context of the whole protein? How the addition of SUMO, at the N-term vs at K120 modifies the 3D structure of the protein?

The authors have mentioned the used of AlphaFold for analyzing SIMs. Could this or other in silico approach be used for the above asked comparison?

If the requested structural comparison has not or cannot be done, at least I consider this is a conceptual topic that fully requires to be included along the Discussion section, also referring to other studies in which the strategy of SUMO fusion have been used or discussed.

In my modest opinion, this last piece of data does not add any significant information to the already strong built story. If I were the one to take the final decision, I would remove it from the manuscript. However, I leave it up to the Editor's decision, which will be based also in the opinion of the other reviewers, whether to keep it or remove it for the final version of the paper.

As requested by the reviewer, we have utilized AlphaFold to predict the structure of the SUMO-fused DHX9^{K120R} in comparison with DHX9. The predicted structure of human DHX9 (AF-Q08211-F1-model_v4) was retrieved from AlphaFold (<https://alphafold.ebi.ac.uk/>) and is presented in **Fig. R1a**. Using the DHX9 structure as a reference, we employed ColabFold to predict the structure of SUMO2-DHX9^{K120R} (Mirdita et al., Nat Methods 2022), shown in **Fig. R1b**. We then aligned the two structures using PyMol software, as depicted in **Fig. R1c**.

Studies of protein structure involving human DHX9 and its fruit fly ortholog, MLE, have indicated that the N-terminal regions (dsRBD1+L1 linker) of both proteins are highly flexible and poorly resolved (Lee et al., Acta Crystallogr D Struct Biol 2023; Jagtap et al., Mol Cell 2023). This disorder is evident in the predicted structures shown in **Figures R1a** and **R1b**. As shown in **Fig. R1a**, K120 is exposed on the surface of the protein structure, even though it is hundreds of amino acids away from the N-terminus of DHX9. Since lysine 120 protrudes toward the dsRBD1 direction, we speculate that native conjugated polySUMO2/3 at K120 can be flexible and move around through the dsRBD1-L1 loop as indicated by the dashed red circle (**Fig. R1a**). Given the high flexibility of the L1 linker and the loop connecting SUMO2 to dsRBD1, we speculate that SUMO2, when fused to the N-terminus of DHX9 K120R, may move freely and transiently mimic the native SUMO conjugation at K120 of WT DHX9 (**Fig. R1c**). Following the reviewer's suggestions, we've now incorporated a description of the SUMO-fusion strategy, along with relevant references, into the results section for Figure 8. Additionally, we've revised the discussion to address this concept. Since the complete structure of DHX9 remains unresolved, we decided not to include the AI-predicted protein structures in the revised manuscript.

Fig. R1 Comparison of protein structures between DHX9^{WT} and 1xSUMO2-DHX9^{K120R}. **a** The AI-assisted prediction of the DHX9 structure was retrieved from AlphaFold (AF-Q08211-F1-model_v4). **b** Using the DHX9 structure as a reference, the structure of SUMO2-DHX9^{K120R} was predicted with the ColabFold. **c** The alignment of the structures from (a) and (b)

11) Discussion:

line 370: please revise expressions such as “Lack of DHX9 SUMOylation”... As already mentioned, we can only assure that the K120R or E122A mutants have diminished SUMOylation or eventually, that they lack SUMOylation at K residue 120, but is not equivalent to say that the DHX9 mutants used in this study entirely lack SUMO conjugation.

Even the authors mentioned few lines below along the discussion section (line 378) that “the K120R mutation in DHX9 doesn't eliminate all SUMO conjugation signals, hinting at additional modification sites”. Therefore, keep this concept clear along the whole manuscript.

We thank the reviewer for pointing this out. We have revised that sentence.

References

Aguilar-Martinez E, Chen X, Webber A, Mould AP, Seifert A, Hay RT, Sharrocks AD. Screen for multi-SUMO-binding proteins reveals a multi-SIM-binding mechanism for recruitment of the transcriptional regulator ZMYM2 to chromatin. *Proc Natl Acad Sci U S A*. 2015 Sep 1;112(35):E4854-63.

Barroso-Gomila O, Trulsson F, Muratore V, Canosa I, Merino-Cacho L, Cortazar AR, Pérez C, Azkargorta M, Iloro I, Carracedo A, Aransay AM, Elortza F, Mayor U, Vertegaal ACO, Barrio R, Sutherland JD. Identification of proximal SUMO-dependent interactors using SUMO-ID. *Nat Commun.* 2021 Nov 18;12(1):6671.

Jagtap PKA, Müller M, Kiss AE, Thomae AW, Lapouge K, Beck M, Becker PB, Hennig J. Structural basis of RNA-induced autoregulation of the DExH-type RNA helicase maleless. *Mol Cell.* 2023 Dec 7;83(23):4318-4333.e10.

Keusekotten K, Bade VN, Meyer-Teschendorf K, Sriramachandran AM, Fischer-Schrader K, Krause A, Horst C, Schwarz G, Hofmann K, Dohmen RJ, Praefcke GJ. Multivalent interactions of the SUMO-interaction motifs in RING finger protein 4 determine the specificity for chains of the SUMO. *Biochem J.* 2014 Jan 1;457(1):207-14.

Lee YT, Sickmier EA, Grigoriu S, Castro J, Boriack-Sjodin PA. Crystal structures of the DExH-box RNA helicase DHX9. *Acta Crystallogr D Struct Biol.* 2023 Nov 1;79(Pt 11):980-991.

Maison C, Bailly D, Roche D, Montes de Oca R, Probst AV, Vassias I, Dingli F, Lombard B, Loew D, Quivy JP, Almouzni G. SUMOylation promotes de novo targeting of HP1 α to pericentric heterochromatin. *Nat Genet.* 2011 Mar;43(3):220-7.

Mirdita M, Schütze K, Moriwaki Y, Heo L, Ovchinnikov S, Steinegger M. ColabFold: making protein folding accessible to all. *Nat Methods.* 2022 Jun;19(6):679-682.

Wen D, Wu J, Wang L, Fu Z. SUMOylation Promotes Nuclear Import and Stabilization of Polo-like Kinase 1 to Support Its Mitotic Function. *Cell Rep.* 2017 Nov 21;21(8):2147-2159.

Wu CS, Ouyang J, Mori E, Nguyen HD, Maréchal A, Hallet A, Chen DJ, Zou L. SUMOylation of ATRIP potentiates DNA damage signaling by boosting multiple protein interactions in the ATR pathway. *Genes Dev.* 2014 Jul 1;28(13):1472-84.

Xu Y, Plechanovová A, Simpson P, Marchant J, Leidecker O, Kraatz S, Hay RT, Matthews SJ. Structural insight into SUMO chain recognition and manipulation by the ubiquitin ligase RNF4. *Nat Commun.* 2014 Jun 27;5:4217.

Zhang T, Yang H, Zhou Z, Bai Y, Wang J, Wang W. Crosstalk between SUMOylation and ubiquitylation controls DNA end resection by maintaining MRE11 homeostasis on chromatin. *Nat Commun.* 2022 Sep 1;13(1):5133.

Zhu S, Hou J, Gao H, Hu Q, Kloeber JA, Huang J, Zhao F, Zhou Q, Luo K, Wu Z, Tu X, Yin P, Lou Z. SUMOylation of HNRNPA2B1 modulates RPA dynamics during unperturbed replication and genotoxic stress responses. *Mol Cell.* 2023 Feb 16;83(4):539-555.e7.

REVIEWER COMMENTS

Reviewer #1 (Remarks to the Author):

In the revised manuscript, the authors did not provide any experimental evidence to address my critiques raised during the previous round of review. While I am ok to compromise and agree with some of the responses, I still feel it is necessary to address whether sumoylation affects DHX9's helicase activity on R-loop resolution in vitro and in cells. Specifically, they need to 1) compare the biochemical activity of DHX9 with/without sumoylation, and 2) examine the level of R-loops either at the global level (by Dot-blot) or at specific gene locus (by DRIP-qPCR) in cells expressing WT, K120R, and SUMO2-K120R DHX9 constructs.

It might be challenging to purify full-length DHX9 for crystallography, but many others have purified DHX9 from insect cells or mammalian cells for biochemical assays. See these references:

1. <https://pubmed.ncbi.nlm.nih.gov/37625785/>
2. <https://pubmed.ncbi.nlm.nih.gov/38110032/>
3. <https://pubmed.ncbi.nlm.nih.gov/34329467/>

Reviewer #3 (Remarks to the Author):

The authors have taken care of all my remaining concerns and suggestions. Overall, I appreciate they have done a great effort to address all the reviewers comments, not only experimentally but also in terms of the articulation and communication of ideas and concepts.

I think the paper has been greatly improved.

I deeply apologise for my delayed revision of this last version.

Point-by-point response to the reviewer's comments

REVIEWER COMMENTS

Reviewer #1 (Remarks to the Author)

In the revised manuscript, the authors did not provide any experimental evidence to address my critiques raised during the previous round of review. While I am ok to compromise and agree with some of the responses, I still feel it is necessary to address whether sumoylation affects DHX9's helicase activity on R-loop resolution in vitro and in cells. Specifically, they need to 1) compare the biochemical activity of DHX9 with/without sumoylation, and 2) examine the level of R-loops either at the global level (by Dot-blot) or at specific gene locus (by DRIP-qPCR) in cells expressing WT, K120R, and SUMO2-K120R DHX9 constructs.

It might be challenging to purify full-length DHX9 for crystallography, but many others have purified DHX9 from insect cells or mammalian cells for biochemical assays. See these references:

1. <https://pubmed.ncbi.nlm.nih.gov/37625785/>
2. <https://pubmed.ncbi.nlm.nih.gov/38110032/>
3. <https://pubmed.ncbi.nlm.nih.gov/34329467/>

We thank the reviewer for her/his constructive comments to improve our manuscript. We agree with the reviewer's points and appreciate the references she/he provided.

Following the reviewer's suggestion, we conducted a helicase assay to compare the biochemical activity of DHX9 with and without SUMOylation. We adapted methods from previous studies (Refs. 59 and 70) to examine the effects of SFB-DHX9-SUMOylation and SFB-DHX9 K120R on the unwinding of DNA/RNA hybrids (rebuttal Fig. R1; **new Supplementary Fig. 6a**). Compared with *in vitro* SUMOylated SFB-DHX9^{WT}, the SFB-DHX9^{K120R} showed no significant difference in the unwinding assay. These new results from the *in vitro* biochemical assay are consistent with and support previous studies, indicating that the N-terminal region of DHX9 is dispensable for its helicase function.

Fig. R1 Comparison of the unwinding activity of WT and mutant SFB-DHX9 (new Supplementary Fig. 6a). Purified SFB-DHX9^{WT} and SFB-DHX9^{K120R} were used in the helicase assay with DNA30-FAM/RNA60 as the substrate. A representative Western blot (left) shows the SUMOylation of WT SFB-DHX9. The helicase activity of purified SFB-DHX9 variants was assessed by gel electrophoresis and fluorescence imaging (middle). The ratios of DNA30-FAM versus the total FAM signal in each group were quantified from three experiments (right).

Ref.

59. Yuan W, et al. TDRD3 promotes DHX9 chromatin recruitment and R-loop resolution. *Nucleic Acids Res* **49**, 8573-8591 (2021).

70. Lee CG, Hurwitz J. A new RNA helicase isolated from HeLa cells that catalytically translocates in the 3' to 5' direction. *J Biol Chem* **267**, 4398-4407 (1992).

Following the reviewer's request, we examined the levels of R-loops at different gene loci using DRIP-qPCR (rebuttal Fig. R2; **new Fig. 8c, and Supplementary Fig. 8c and 8d**). Unlike DHX9^{K120R}, SUMO2-fused DHX9^{K120R} (S2-K120R) reduces R-loop levels at the examined gene regions to levels similar to those of WT DHX9. These new observations further strengthen our findings in Fig. 8b and 8d, showing that S2-K120R not only restores association with R-loops but also alleviates the survival defects associated with DHX9^{K120R}.

Fig. R2 SUMO2-DHX9^{K120R} (S2-K120R) restores the normal level of R-loops (new Fig. 8c, and Supplementary Fig. 8c and 8d).

Left (new Fig. 8c): The relative R-loop level at the 5' pause region of β -actin gene was determined by RT-qPCR. The representative bar graph was from three separate experiments. Values are normalized to β -actin in1 and presented as mean \pm SEM.

Middle (new Supplementary Fig. 8c): The relative R-loop level at the B region of γ -actin gene was determined by RT-qPCR. The representative bar graph was from three separate experiments. Values are normalized to γ -actin in1 and presented as mean \pm SEM.

Right (new Supplementary Fig. 8d): Duplicated samples from Fig. 8c and Supplementary Fig. 8c were used to confirm the protein level of SFB-DHX9 variants and SUMO2-DHX9^{K120R} by Western blot with anti-DHX9 antibody.

Reviewer #3 (Remarks to the Author)

The authors have taken care of all my remaining concerns and suggestions. Overall, I appreciate they have done a great effort to address all the reviewers comments, not only experimentally but also in terms of the articulation and communication of ideas and concepts.

I think the paper has been greatly improved.

I deeply apologise for my delayed revision of this last version.

We thank this reviewer for her/his time reevaluating our revisions. We appreciate this reviewer for making several constructive comments to improve our manuscript. We are very pleased that the reviewer is satisfied with the quality of our manuscript.

REVIEWERS' COMMENTS

Reviewer #1 (Remarks to the Author):

In the revised manuscript, the authors have addressed my concerns, and the conclusions are now fully supported by the results. Therefore, I recommend accepting the manuscript for publication.

Point-by-point response to the reviewer's comments

REVIEWER COMMENTS

Reviewer #1 (Remarks to the Author)

In the revised manuscript, the authors have addressed my concerns, and the conclusions are now fully supported by the results. Therefore, I recommend accepting the manuscript for publication.

We thank this reviewer for taking the time to reevaluate our revised manuscript. We sincerely appreciate the reviewer for making several constructive comments. We are very pleased that the reviewer supports the publication of our manuscript.